# CD11b⁺ lung dendritic cells at different stages of maturation induce Th17 or Th2 differentiation

Gentaro Izumi[1,3,4], Hideki Nakano [1,4 ✉], Keiko Nakano [1], Gregory S. Whitehead[1], Sara A. Grimm [2], Michael B. Fessler [1], Peer W. Karmaus[1] & Donald N. Cook[1 ✉]

Dendritic cells (DC) in the lung that induce Th17 differentiation remain incompletely understood, in part because conventional CD11b⁺ DCs (cDC2) are heterogeneous. Here, we report a population of cDCs that rapidly accumulates in lungs of mice following house dust extract inhalation. These cells are Ly-6C⁺, are developmentally and phenotypically similar to cDC2, and strongly promote Th17 differentiation ex vivo. Single cell RNA-sequencing (scRNA-Seq) of lung cDC2 indicates 5 distinct clusters. Pseudotime analysis of scRNA-Seq data and adoptive transfer experiments with purified cDC2 subpopulations suggest stepwise developmental progression of immature Ly-6C⁺Ly-6A/E⁺ cDC2 to mature Ly-6C⁻CD301b⁺ lung resident cDC2 lacking *Ccr7* expression, which then further mature into CD200⁺ migratory cDC2 expressing *Ccr7*. Partially mature Ly-6C⁺Ly-6A/E⁻CD301b⁻ cDC2, which express *Il1b*, promote Th17 differentiation. By contrast, CD200⁺ mature cDC2 strongly induce Th2, but not Th17, differentiation. Thus, Th17 and Th2 differentiation are promoted by lung cDC2 at distinct stages of maturation.

[1] Immunity, Inflammation and Disease Laboratory, Division of Intramural Research, National Institute of Environmental Health Sciences, NIH, Research Triangle Park, NC, USA. [2] Integrative Bioinformatics Support Group, Division of Intramural Research, National Institute of Environmental Health Sciences, NIH, Research Triangle Park, NC, USA. [3] Present address: Department of Obstetrics and Gynecology, Faculty of Medicine, University of Tokyo, Hongo, Bunkyo-ku, Tokyo, Japan. [4] These authors contributed equally: Gentaro Izumi, Hideki Nakano. ✉email: nakanoh@niehs.nih.gov; cookd@niehs.nih.gov

Th17 cells are critical mediators of mucosal host defense. Strategically positioned at barrier sites, such as the intestine, skin, and lung, these IL-17-producing CD4[+] T cells have non-redundant functions in protection against multiple extracellular pathogens. However, dysregulation of Th17 cell activity is also central to many widespread pulmonary diseases, including cystic fibrosis, chronic obstructive pulmonary disease, and asthma[1]. Almost half of asthmatics are resistant to steroids and these individuals often present with neutrophilic inflammation of the airway[2,3]. Th17 cells likely contribute to this form of asthma, as disease severity correlates with the number of circulating CD4[+]IL-17[+] cells[4–6]. Furthermore, Th17 cells are resistant to glucocorticoids[7], which might account, at least in part, for the observation that neutrophilic asthma is often steroid-resistant. Improved mechanistic understanding of how these cells develop might lead to improved strategies to reduce the incidence of neutrophilic asthma and other Th17-dependent pulmonary diseases.

Differentiation of naive CD4[+] T cells to effector cells, including Th17 and Th2 cells, is driven by dendritic cells (DCs)[8,9]. Conventional DCs (cDCs) are derived exclusively from FMS-like tyrosine kinase 3 ligand (FLT3L)-dependent DC precursors (preDCs)[9–11], and are thus developmentally distinct from monocyte-derived cells, which arise independently of FLT3L. Monocyte-derived cells can be phenotypically identified by high levels of certain cell surface markers, including Fc-gamma receptor FcγRI/CD64 and the complement receptor C5aR1/CD88 in human and mouse, and F4/80 in mouse[12–14]. In the mouse lung, cDCs comprise two major subsets that are often defined by their reciprocal display of two integrins, CD11b and CD103[9,15]. cDCs expressing high amounts of CD103 and relatively low amounts of CD11b are a homogeneous population developmentally dependent on the transcription factor, BATF3, and referred to as CD103[+] cDCs, or simply "DC1"[16,17]. Human cDC1 can be identified by their display of CD141 on their surface. Although cDC1 can stimulate CD4[+] T cells, these cDCs are best known for their ability to cross-present antigens to CD8[+] T cells[18–20]. The second major population of mouse lung cDCs display high levels of CD11b, but low amounts of CD103, and are thus called CD11b[+] cDCs, or "cDC2". Human cDC2 are defined by their cell surface display of CD1d. cDC2 are largely dependent on interferon regulatory factor 4 (IRF4) for their development[21], and efficiently promote the differentiation of naive CD4[+] T cells to effector helper T cells[22,23]. Unlike cDC1, cDC2 are very heterogenous[20,24,25], and this has confounded analyses of their function. Accordingly, the major cDC subset in the lung that drives Th17 responses to inhaled antigens remains uncertain, with both cDC1[18] and IRF4-dependent cDC2[22] being implicated in different studies. cDC2 can also induce Th2 cells[22,23], although it is unclear whether the same subpopulation of cDC2 induces both Th2 and Th17 cells[22]. In vitro, IL-1β, IL-6, and TGF-β promote Th17 differentiation[26,27]. However, it is unknown whether a specific cDC2 subpopulation produces these factors, in part because cell surface markers that reliably distinguish between different types of cDC2 are poorly defined. It is well established that T cell differentiation is induced in the tissue-draining lymph nodes (LNs)[28], but lung cDC2 are less migratory than cDC1[29], suggesting that some effector T cells might also be induced by lung-resident cDCs.

In this work, we utilize mass cytometry, single-cell RNA sequencing (scRNA-Seq), and ex vivo studies to identify and functionally analyze discrete cDC2s in the lungs of mice following house-dust extract (HDE) inhalation. These cDC2s resolve into five distinct clusters that differ in their maturational status and ability to induce helper T cell differentiation. Whereas partially mature Ly-6C[+]CD301b[−] cDC2 lack Ccr7 expression and promote Th17 differentiation, more mature CD200[+] cDC2 express Ccr7 and induce Th2, but not Th17, differentiation.

## Results

### HDE induces Ly-6C[+] cDC2-like cell accumulation in the lung.
Previous studies have shown that when inhaled together with ovalbumin (OVA), adjuvants such as lipopolysaccharide (LPS)[30] or HDEs[31,32], can promote the in vivo development of OVA-specific Th17 cells that in turn drive airway neutrophilia upon subsequent exposure to that same protein. In agreement with this, levels of IL-17 and IL-13, as well as numbers of neutrophils and eosinophils, were significantly increased in the airways of mice after sensitization with HDE/OVA and challenge with OVA aerosol (Fig. 1a, b and Supplementary Fig. 1a, b). Helper T cell differentiation is known to be induced in the tissue-draining LNs[28], but it remains unclear whether this can also occur in the lung. To investigate whether Th17 differentiation can occur in the lung, we measured IL-17 in the lung and mediastinal LNs (mLNs) at various times post-HDE/OVA sensitization. In parallel, we measured IL-13 to assess Th2 differentiation. By 2 days post-sensitization, IL-13 was elevated in mLNs, but did not appear in the lung until 4 days sensitization (Supplementary Fig. 1c). This suggests Th2 cells arise in mLNs and subsequently migrate to the lung. By contrast, IL-17 was elevated in both the lung and mLNs by 2 days post-sensitization (Supplementary Fig. 1c), suggesting that Th17 cells can simultaneously develop in both locations. Taken together, these results suggest that lung-resident cDCs can promote the development of allergen-specific Th17 cells. In support of this, total lung DCs isolated from HDE/OVA-treated mice primed the development of IL-17-producing Th17 cells, as well as IL-13-producing Th2 cells (Fig. 1c), in agreement with a previous report[33].

To investigate the profiles of lung cDC populations, we used mass cytometry to compare lung cDCs at steady state and following HDE/OVA allergic sensitization. Unsupervised analysis revealed two cell populations within the CD45[+] leukocyte gate that were present only in HDE/OVA-treated mice (Fig. 1d). tSNE analysis identified one of those two populations as interstitial macrophages (IMs), and the other as cDC2. A more restrictive analysis of mononuclear cells (CD45[+]CD3ε[−]CD19[−]NK1.1[−]Ly-6G[−]) (Supplementary Fig. 2a) showed that the second population of HDE-induced cells was similar to, but distinct from, traditional cDC2 (Fig. 1d and Supplementary Fig. 2b). These cells displayed low amounts of the macrophage markers, F4/80, CD88, and Siglec-F, but displayed high levels of Ly-6C (Fig. 1e), a marker usually associated with inflammatory monocytes[20]. This display of Ly-6C allowed us to use flow cytometry to distinguish the HDE-induced cDC2-like cells from conventional cDC2 (Fig. 1f). To evaluate the cDC2-like Ly-6C[+] cells by flow cytometry, we gated for cDCs (CD45[+]CD11c[+]MHC-II[+]CD88[−]F4/80[−]Siglec-F[−]), excluding Siglec-F[+] alveolar macrophages (AMs), CD88[+]F4/80[+] IMs, F4/80[+] monocytes, and CD88[+]MHC-II[−] neutrophils (Supplementary Figs. 1d and 3a, b). At steady state, there were very few Ly-6C[+] antigen-presenting cells (APCs), but their numbers increased dramatically by 12 h post-treatment, peaked between 18 and 24 h, then declined rapidly and stabilized by 48 h (Fig. 1f, g). cDCs having low display of Ly-6C (Ly-6C[−] cDC2) were abundant at steady state, but their numbers also increased by 24 h post-HDE/OVA inhalation, and returned to baseline by 48 h (Fig. 1g). Number of monocytes and IMs also increased after HDE/OVA inhalation, whereas those of cDC1 and AMs were much less affected by this treatment (Fig. 1g and Supplementary Fig. 3c).

### Ly-6C[+] HDE-induced APCs are cDCs.
Like cDC1 and cDC2, HDE-induced Ly-6C[+] APCs also formed dendrites typical of DCs (Fig. 2a), but their high display levels of Ly-6C prompted us to

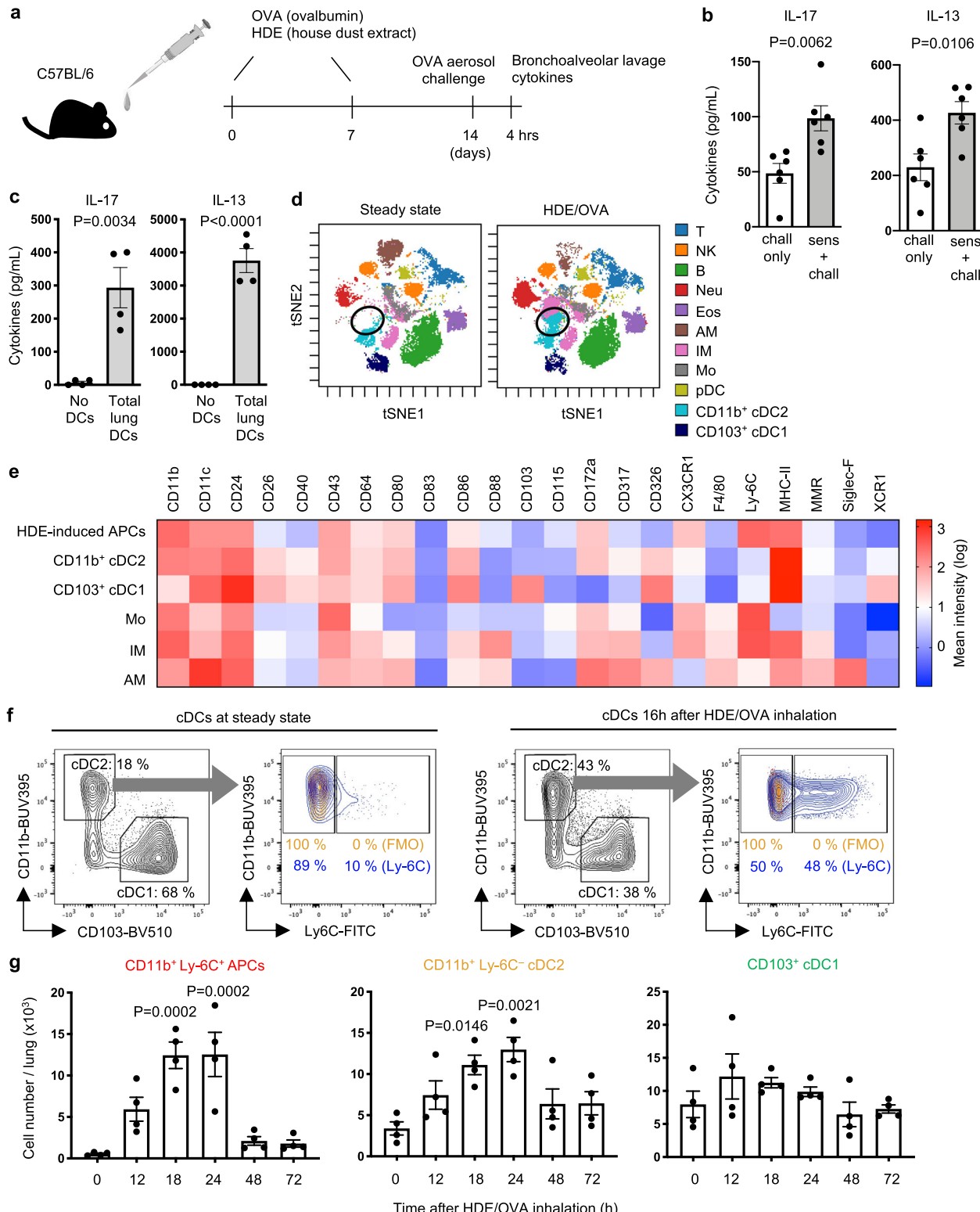

test whether these HDE-induced APCs were indeed bona fide cDCs, or were instead derived from monocytes. We purified multiple APC subsets and from the lung, prepared RNA from them, and performed bulk RNA-sequencing (RNA-Seq). Principal component analysis (PCA) and differentially expressed gene (DEG) analysis revealed that the transcriptional profile of Ly-6C+ HDE-induced APCs was very similar to that of cDC2 (relatively few DEGs), and more distantly related to the transcriptomes of

other APCs (more DEGs), including cDC1s, preDCs and monocytes (Fig. 2b, c and Supplementary Fig. 4a). For example, the cDC signature genes, *Dpp4, H2eb1, Itgax, Kmo*, and *Zbtb46*[13,34,35,13,36] were expressed in Ly-6C+ HDE-induced APCs and in Ly-6C− cDC2 and cDC1, but not in monocytes (Fig. 2d and Supplementary Fig. 4b). Conversely, expression of macrophage-signature genes[36,37] was lower in Ly-6C+ HDE-induced APCs than in monocytes (Supplementary Fig. 4b).

**Fig. 1 Accumulation of Ly-6C$^+$ cDC2-like cells in the lung following allergic sensitization. a** Timeline for allergic sensitization and allergen challenge in HDE/OVA-mediated model of asthma. **b** Cytokine production in lungs of challenged mice, as measured by ELISA. Data were analyzed by unpaired two-tailed $t$-test ($n = 6$). Data are presented as mean ± SEM. A representative result from two independent experiments is shown. **c** Cytokines produced by OT-II CD4$^+$ T cells stimulated by total lung DCs isolated from HDE/OVA-treated C57BL/6 mice. Data are presented as mean ± SEM. Data were analyzed by unpaired two-tailed $t$-test ($n = 4$). The gating strategy for DC sorting is shown in Supplementary Fig. 3a. Data are from a single experiment, representative of three. **d** tSNE plots showing mass cytometry analysis of lung leukocytes from untreated mice and at 16 h post-HDE/OVA inhalation. T: T cells, NK: natural killer cells, B: B cells, Neu: neutrophils, Eos: eosinophils, AM: alveolar macrophages, IM: interstitial macrophages, Mo: monocytes, pDC: plasmacytoid DCs. Black circles indicate cDC2-like cells seen only in HDE/OVA-treated mice. **e** Heatmap of surface marker levels on cells analyzed by mass cytometry. A representative result from two independent experiments is shown. **f** Cytograms for flow cytometric analysis of lung cDCs at steady state and 16 h post-HDE/OVA inhalation. cDCs are CD45$^+$CD11c$^+$I-A$^+$CD88$^-$Siglec-F$^-$F4/80$^-$Live/Dead$^-$. Further gating information is shown in Supplementary Fig. 1d. Blue or orange contour plots show cells stained with anti-Ly-6C Abs or fluorescence minus one (FMO), respectively. **g** Time course for the number of cDCs and Ly-6C$^+$ APCs following HDE/OVA inhalation. The gating strategy is shown in Supplementary Fig. 3b. Data were analyzed by ordinary one-way ANOVA with Tukey's multiple comparison test ($n = 4$). Data are presented as mean ± SEM. A representative result from two independent experiments is shown. Source data are provided as a Source Data file.

However, the cDC-associated genes, *Dpp4* and *Zbtb46*, were not as highly expressed in Ly-6C$^+$ HDE-induced APCs as in the other cDC subsets. Given that Ly-6C is also displayed on preDCs in the lung and bone marrow (BM) (Supplementary Fig. 5c)[38], we reasoned that Ly-6C$^+$ APCs might be immature cDCs. In agreement with this, accumulation of Ly-6C$^+$ APCs in the lungs of HDE-treated animals was severely impaired in *Ccr2$^{-/-}$ Cx3cr1$^{-/-}$* double knockout (DKO) mice (Supplementary Fig. 5d), reminiscent of a similar finding for preDCs[38].

*Zbtb46* is exclusively expressed by cDCs[34,35], and the promoter of this gene has been used to express diphtheria toxin receptor (DTR) transgenes, thereby conferring selective sensitivity to diphtheria toxin (DTX) in cDCs. As expected, administration of DTX to these *Zbtb46-DTR* transgenic mice significantly decreased the number of cDC1 and cDC2, and also decreased Ly-6C$^+$ HDE-induced APCs (Fig. 2e). By contrast, monocytes, AMs, and IMs were not decreased. Similarly, *Ftl3l$^{-/-}$* mice lacking FLT3L had significantly fewer cDC1, cDC2, and Ly-6C$^+$ HDE-induced APCs compared with wild-type (WT) mice (Fig. 2f). Together, these observations suggest that the Ly-6C$^+$CD11b$^+$F4/80$^-$CD88$^-$Siglec-F$^-$ APCs induced by HDE inhalation are bona fide cDCs, and we henceforth refer to them as "Ly-6C$^+$ cDC2".

**Ly-6C$^+$ cDC2 promote Th17 cell differentiation.** All cDC subsets, including Ly-6C$^+$ cDC2, took up fluorescently labeled OVA upon its instillation into the airways of mice (Fig. 3a, b). The antigen processing ability of cDCs was also analyzed using DQ-OVA, which fluoresces upon its digestion by intracellular proteases. The frequency of OVA$^+$Ly-6C$^+$ cDC2 and their mean fluorescent intensity (MFI) were either similar to, or greater than, the corresponding values for the other cDC subsets (Supplementary Fig. 7a, b), suggesting that Ly-6C$^+$ cDC2 have the potential to capture antigens, degrade them, and present antigen-derived peptides to T cells.

We next examined the expression of *Il1b*, *Il6*, and *Tgfb1*, which encode cytokines that promote Th17 differentiation[26,27,39]. The RNA-Seq data revealed that each of these three genes was more highly expressed in Ly-6C$^+$ cDC2 than in either Ly-6C$^-$ cDC2 or cDC1 (Fig. 3c). Ly-6C$^+$ cDC2 also expressed high levels of *Casp1* and *Casp4/11*, whose encoded proteins activate IL-1β (Supplementary Fig. 7c)[40]. *Il23a*, which encodes the IL-23A subunit that supports the survival of Th17 cells[41], was highly expressed by Ly-6C$^-$ cDC2. By contrast, *Il12b*, which encodes the p40 subunit of the Th1-promoting cytokine IL-12, was lower in Ly-6C$^+$ cDCs2 than in other cDC subsets (Fig. 3c).

To directly test the Th17 cell-inducing ability of Ly-6C$^+$ cDC2, we purified these cells and co-cultured them with naive OVA-specific CD4$^+$ T cells from OT-II TCR transgenic mice. cDC1 and Ly-6C$^-$ cDC2 were also tested for comparison. Compared with other cDC subsets, Ly-6C$^+$ cDC2 induced only moderate proliferation of T cells (Fig. 3d). Consistent with this moderate proliferation, IL-2 production in the co-culture of T cells with Ly-6C$^+$ cDC2 was lower than in co-cultures containing other cDC subsets (Supplementary Fig. 7d). The lower IL-2 production and moderate proliferation induced by Ly-6C$^+$ cDC2 might be due to lower levels of MHC class II and co-stimulatory molecule CD86 on their surface compared with other cDC subsets (Supplementary Fig. 7e, f), in agreement with our results from mass cytometry analysis (Fig. 1e). Strikingly, however, Ly-6C$^+$ cDC2 strongly promoted Th17 differentiation. By comparison, Ly-6C$^-$ cDC2 and cDC1 had only weak activity in this regard (Fig. 3d and Supplementary Fig. 7g). Neutralizing antibody (Ab)-mediated blockade of either IL-1β (Fig. 3e and Supplementary Fig. 7h) or IL-6 (Fig. 3f and Supplementary Fig. 7i) markedly suppressed Ly-6C$^+$ cDC2-directed Th17 differentiation, while having either no effect or only modest effects on T cell proliferation.

**Ly-6C$^+$ cDC2 promote Th17 response in vivo.** The induction of Th17 differentiation by Ly-6C$^+$ cDC2 ex vivo prompted us to test whether Ly-6C$^+$ cDC2 were sufficient to promote Th17 responses in vivo. To this end, we generated a mouse strain in which Ly-6C$^+$ cDC2 represent the vast majority of cDCs in the lung. We first bred mice carrying a Cre recombinase-inducible gene encoding the diptheria toxin alpha subunit (*DTA$^{fx}$*) with mice carrying a *Cre* gene under control of the *Itgax* (CD11c) promoter (*Itgax$^{Cre}$* x *DTA$^{fx}$*). In these mice, previously termed ΔDC[42], many *Cd11c*-expressing cells, including cDCs, are reported to undergo spontaneous cell death. Our experiments confirmed this, although some cDC1 and Ly-6C$^+$ cDC2 remained (Supplementary Fig. 8a–d). To selectively reduce the number of BATF3-dependent cDC1[17], we crossed ΔDC mice with BATF3-deficient mice and found that the offspring of this cross (*Batf3$^{-/-}$* ΔDC mice) essentially lacked all lung DC subsets except Ly-6C$^+$ cDC2 (Fig. 4a–d). We then studied responses of *Batf3$^{-/-}$* ΔDC mice in an HDE-mediated model of asthma[32] (Fig. 4e). Compared with WT mice, *Batf3$^{-/-}$* ΔDC animals displayed reduced accumulation of IL-13 (Fig. 4f) and eosinophils (Fig. 4g) in the airways, indicating diminished Th2 responses. By contrast, airway neutrophilia in *Batf3$^{-/-}$* ΔDC mice was as high, or higher, than that seen in WT mice (Fig. 4g). Furthermore, IL-17 production from lungs of *Batf3$^{-/-}$* ΔDC mice was comparable, or even higher, than in WT mice (Fig. 4f). IFN-γ production was elevated in *Batf3$^{-/-}$* ΔDC mouse lungs, suggesting that these mutant mice can develop Th1 responses. Together, these results show that the Ly-6C$^+$ cDC2 selectively retained in *Batf3$^{-/-}$* ΔDC mice are sufficient to promote Th17 development, which in turn drives allergen-dependent airway neutrophilia.

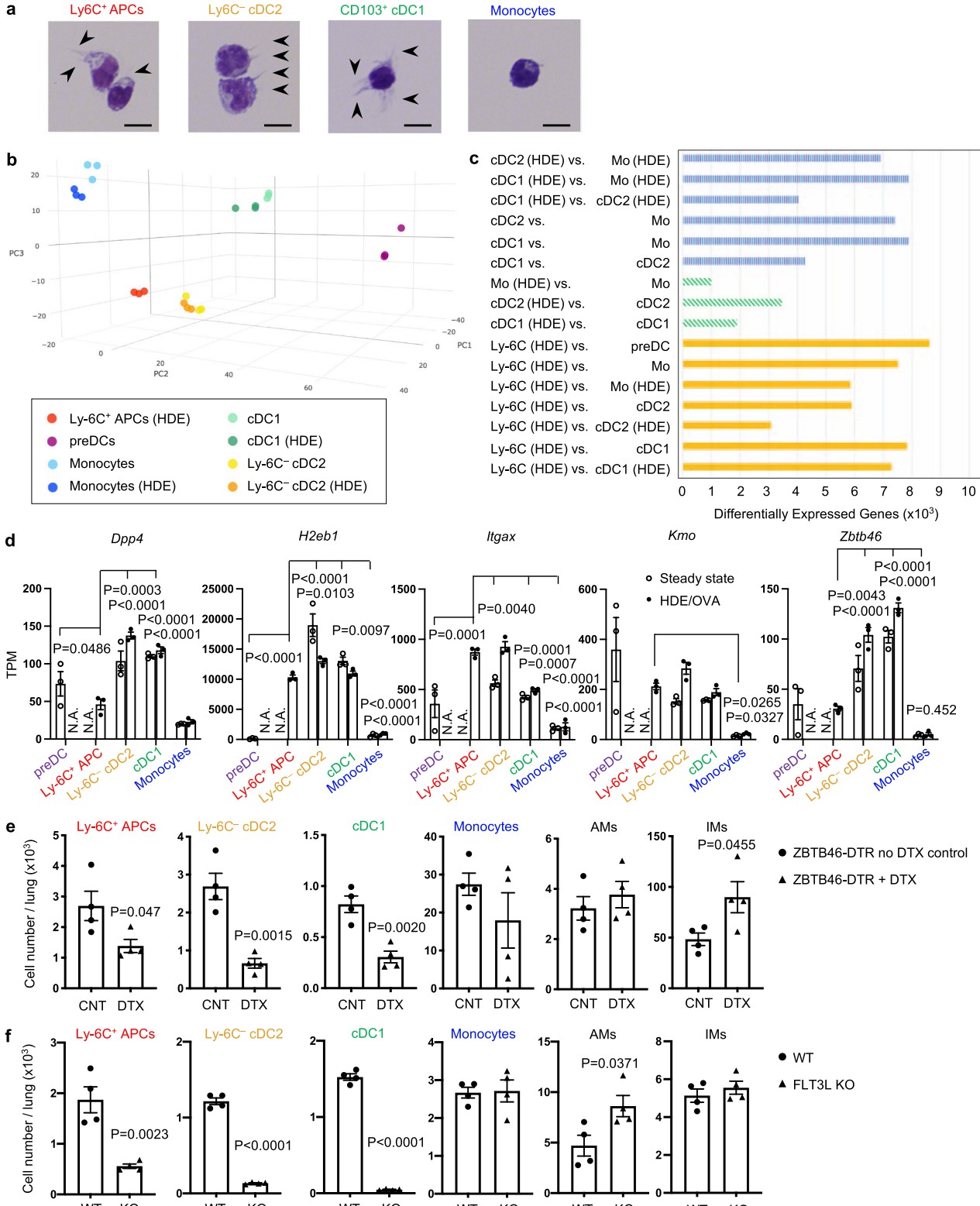

**Fig. 2 Lineage analysis of HDE-induced Ly-6C⁺ APCs. a** Morphology of HDE-induced Ly-6C⁺ APCs, Ly-6C⁻ cDC2, cDC1, and monocytes purified by flow cytometry. Dendrites are indicated by arrowheads. Bars denote 10 μm. A representative result from two independent experiments is shown. **b** Principal component analyses of RNA-Seq data for these same cell types isolated from mouse lungs at steady state and 16 h after HDE/OVA inhalation (HDE). **c** Number of DEGs between the indicated pairs of cell populations. Mo: monocytes. **d** Expression of cDC-signature genes is shown as transcripts per million (TPM) from RNA-Seq analysis. Data were analyzed by two-way ANOVA with Fisher's LSD multiple comparison test ($n = 3$). Data are presented as mean ± SEM. **e**, **f** Cell numbers for indicated cell populations in HDE/OVA-treated *Zbtb46-DTR* mice (C57BL/6 background) with or without DTX treatment (**e**), or HDE/OVA-treated WT or FLT3L KO mice (C57BL/6 background) (**f**), as determined by flow cytometry. The gating strategy is shown in Supplementary Fig. 3b. Data were analyzed by unpaired two-tailed *t*-test ($n = 4$). Data are presented as mean ± SEM. A representative result from two independent experiments is shown. Source data are provided as a Source Data file.

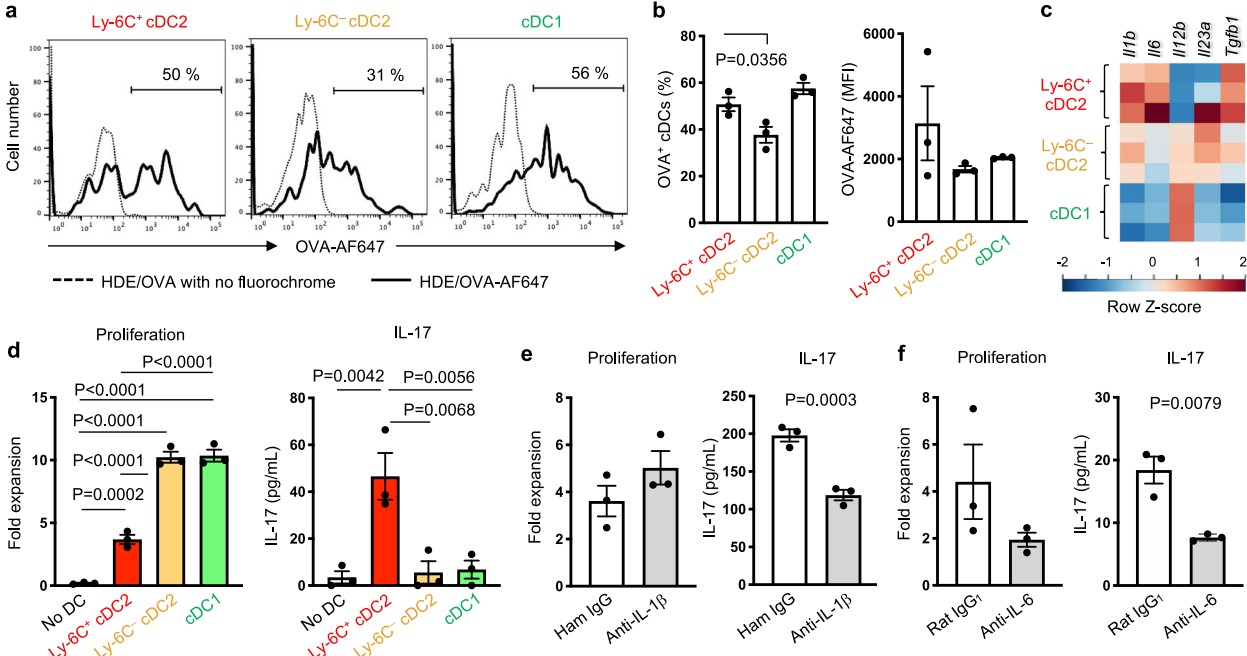

**Fig. 3 Ly-6C+ CD11b+ cDC2 stimulate Th17 cell differentiation. a, b** Flow cytometric analysis of lung cDCs harvested 16 h after inhalation of AF647-labeled OVA with HDE. Representative histograms (**a**) and compiled data ($n = 3$) (**b**) showing OVA-AF647 uptake. The gating strategy is shown in Supplementary Fig. 6a. Data were analyzed by ordinary one-way ANOVA with Dunnett's multiple comparison. Data are presented as mean ± SEM. **c** Heatmap of RNA-Seq data showing expression of genes promoting Th17 or Th1 differentiation. Expression levels are indicated by color difference as shown in bottom bar. **d** Proliferation of and IL-17 production from CD4+ T cells stimulated by lung cDCs. IL-17 in the supernatant of cultured CD4+ T cells was measured by ELISA. Data were analyzed by ordinary one-way ANOVA with Tukey's multiple comparison test ($n = 3$). Data are presented as mean ± SEM. **e, f** Proliferation of and IL-17 production from CD4+ T cells stimulated by Ly-6C+ cDC2 in the presence of anti-IL-1β (**e**) or anti-IL-6 (**f**) neutralizing Abs or isotype control. IL-17 in the supernatant was measured by BioPlex. Data were analyzed by unpaired two-tailed t-test ($n = 3$). Data are presented as mean ± SEM. A representative result from two independent experiments is shown. Source data are provided as a Source Data file.

**Ly-6C+ cDC2 give rise to a subpopulation of Ly-6C− cDC2.** Their short life span (Fig. 1g), gene expression profile (Fig. 2d), and display of Ly-6C (Fig. 1e, f) all suggested that Ly-6C+ cDC2 might represent an immature stage of cDC2. To study this, we purified cDC2 and monocytes from the lung and cultured them in vitro. During culture, monocytes maintained Ly-6C on their surface, whereas Ly-6C+ cDC2 lost Ly-6C, and, at least based on their display of surface markers, became even more similar to Ly-6C− cDC2 (Fig. 5a, b and Supplementary Fig. 9b). The transcriptome of Ly-6C+ cDC2 was more similar to that of Ly-6C− cDC2 than to either cDC1 or monocytes, before and after in vitro culture (Fig. 5c). Nonetheless, cultured Ly-6C+ cDC2 remained transcriptionally distinct from Ly-6C− cDC2 (Fig. 5c and Supplementary Fig. 10a), raising the possibility that Ly-6C+ cDC2 are not simply direct precursors of Ly-6C− cDC2.

Although valuable in many experimental settings, gene profiling of bulk cells has a limited capacity to detect heterogeneity in cell populations, including cDC2. We therefore sorted total cDC2 from the lung following HDE/OVA treatment, and studied them at the single-cell level using scRNA-Seq. Analysis of the data by Seurat software[43] revealed seven distinct clusters within total cDC2 population (Fig. 5d). Examination of genes uniquely expressed by each cluster (Fig. 5e and Supplementary Fig. 10b), as well as comparisons to clusters previously identified in the lungs of naive mice by Han et al.[24] (Fig. 5f) and lungs of virus infected mice by Bosteels et al.[25] (Supplementary Fig. 10c), indicated that cluster 7 is mixture of Xcr1-expressing cDC1s and dividing DCs, whereas cluster 4 is mixture of AMs and IMs. As the latter cells had transcriptional properties of both macrophages and cDC2, they might be equivalent to monocyte-derived DCs recently reported by Menezes et al.[44] By contrast, the remaining five clusters (1, 2, 3, 5, and 6) expressed cDC

signature genes but not macrophage genes, suggesting they are all bona fide cDC2 (Fig. 5e, f and Supplementary Fig. 10b–d).

Since Ly-6C is a surface marker on cDC2 that induce Th17 differentiation (Fig. 3d), we evaluated cell surface Ly-6C by labeling the cells prior to lysis with an oligonucleotide-labeled Ab directed at that protein (indexed scRNA-Seq)[45]. Integration of sequence data corresponding to this bar-coded oligonucleotide together with sequencing data for mRNA allowed us to measure cell surface Ly-6C on each cDC2 cluster. We found that although Ly6c2 RNA is only highly expressed in cluster 6, Ly-6C cell surface protein was present on clusters 1, 3, and 6 (Fig. 6a). This discrepancy between protein and RNA might result from the retention of Ly-6C protein on the cell surface after transcription of the Ly6c2 gene ceased. If so, clusters 1 and 3 may represent more mature forms of cDC2 than cluster 6. The majority of cells in cluster 2 and 5 were negative for both Ly6c2 mRNA and Ly-6C protein (Fig. 6a), suggesting that they might represent relatively mature cDC2.

To analyze the maturation pathway of the above-described cDC2 subpopulations, we needed to first develop a strategy for their purification. We therefore queried the scRNA-Seq data for DEGs encoding cell surface proteins that could be leveraged in flow cytometry experiments to purify cDCs corresponding to the different clusters (Fig. 6b). We found that clusters could be identified using antibodies (Abs) against the following markers: clusters 1 and 3 [Ly-6C+Ly-6A/E−]; cluster 2 [CD200+]; cluster 4 [CD14+]; cluster 5 [CD301b+ (encoded by Mgl2)]; and cluster 6 [Ly-6C+Ly-6A/E+] (Fig. 6c). Using these Abs, we found that number of cDCs within clusters 1, 3, and 6 increased dramatically after HDE/OVA-mediated allergic sensitization, peaked around 18–24 h later, and then declined (Fig. 6d). The accumulation of

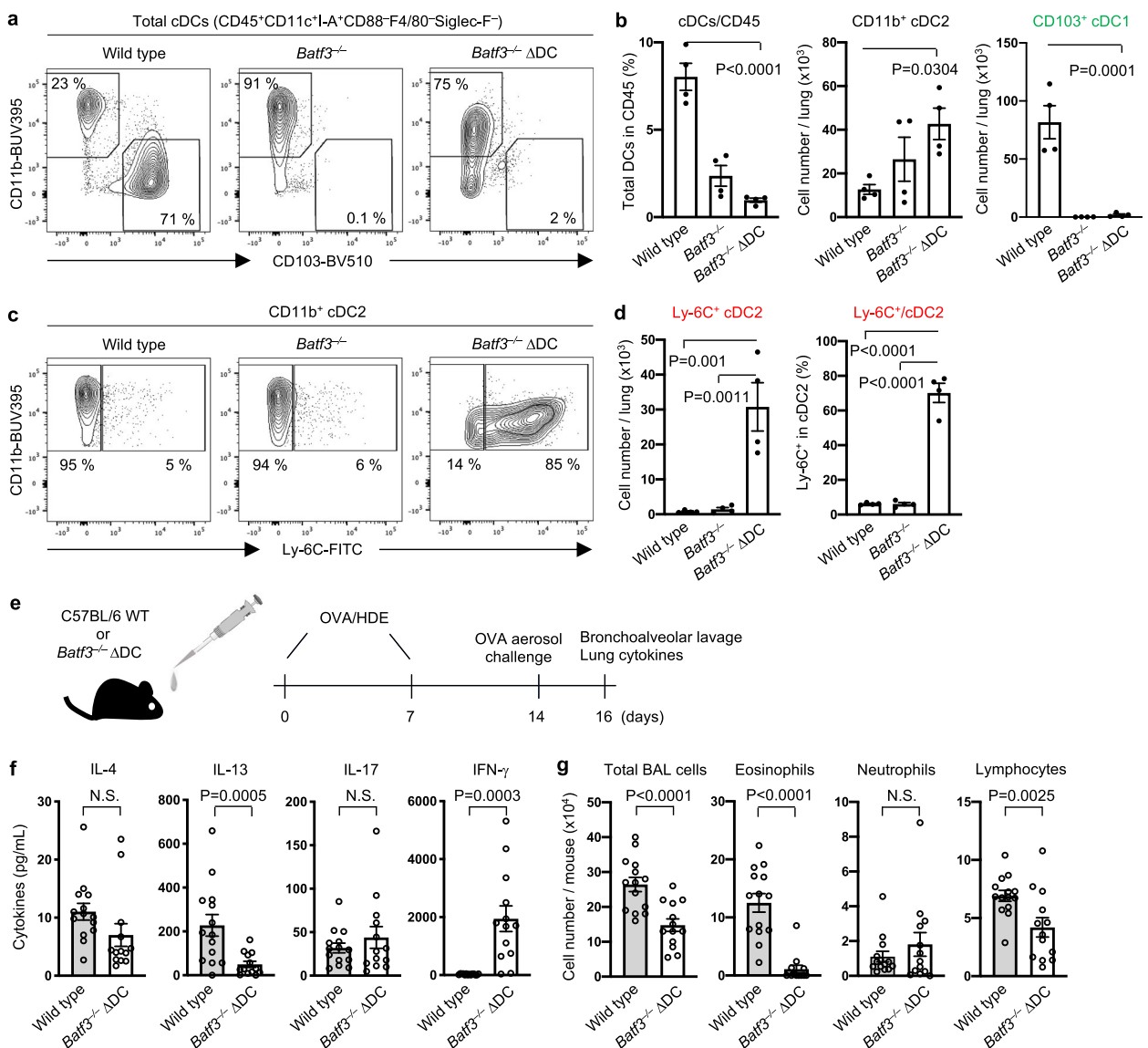

**Fig. 4 Ly-6C⁺ cDC2 are sufficient for induction of Th17-dependent neutrophilic airway inflammation. a–d** Analysis of cDCs (CD45⁺CD11c⁺I-A⁺CD88⁻F4/80⁻Siglec-F⁻Live/Dead⁻) in the lungs of WT C57BL/6, C57BL/6-*Batf3⁻/⁻* and C57BL/6-*Batf3⁻/⁻* ΔDC mice at steady state, including representative cytograms (**a**, **c**), and compiled data (**b**, **d**) showing percentages and cell numbers for total cDC, cDC2, and cDC1 (**b**), and for Ly-6C⁺ cDC2 (**d**). The gating strategy is shown in Supplementary Fig. 1d. Data were analyzed by ordinary one-way ANOVA with Dunnett's multiple comparison test (*n* = 4). Data are presented as mean ± SEM. **e** Timeline for allergic sensitization and allergen challenge in HDE/OVA-mediated mouse model of asthma. **f** Cytokine production in lungs of challenged mice, as measured by ELISA. **g** Cell numbers for the indicated leukocyte subsets in BALF of allergen-challenged WT and *Batf3⁻/⁻* ΔDC mice. Data were analyzed by ordinary one-way ANOVA with Tukey's multiple comparison test (*n* = 14 WT and *n* = 13 *Batf3⁻/⁻* ΔDC mice). Data are presented as mean ± SEM. Combined results of two independent experiments are shown. Source data are provided as a Source Data file.

cDCs within each of these clusters was dependent on the chemokine receptors, CCR2 and CX3CR1 (Supplementary Fig. 12a), suggesting descendance of these cDCs from newly migrated preDCs[38]. Cluster 5 was the major cDC2 population present at steady state, and numbers of these cells did not change dramatically post-sensitization, suggesting they are lung-resident cDC2.

Developmental trajectory (pseudotime) analysis of the clusters using Monocle[46] suggested that cDC2 likely mature in the following order; clusters 6, 1, 3, and 5 (Fig. 7a, b). While cluster 2 might also descend from cluster 3, there is a large gap in pseudotime between those two clusters (Fig. 7a, b), and the transcriptome of cluster 2 is very different from those of the other clusters (Fig. 5e and Supplementary Fig. 10e). Cluster 2 might therefore represent a cDC2 population that is independent of Ly-

6C⁺cDC2. To study the maturation of cDC2 subpopulations in vivo, we isolated Ly-6C⁺CD301b⁻CD200⁻ cDC2 (corresponding to clusters 6, 1, and 3) from lungs of C57BL/6J (CD45.2) mice and adoptively transferred these cells to CD45.1 recipients (Fig. 7c). Analysis of donor-cDC2-derived CD45.2⁺ cells recovered from recipient mice revealed the transferred cells had lost Ly-6C, and had slightly increased CD200, but not CD301b, by 1 day post-transfer (Fig. 7d and Supplementary Fig. 12b). However, by 3 days post-transfer, some donor cDC2 had gained CD301b, whereas CD200 levels were unchanged compared with cells harvested at day 1. These data suggest that CD301b⁺ cells in cluster 5 descend from Ly-6C⁺ cells in clusters 6, 1, and 3.

Cluster 2 is a minor population in the lung at steady state (equivalent to cluster #29 in the study of Han et al.)[24] (Fig. 5f), but number of these cells increased dramatically after HDE/OVA-

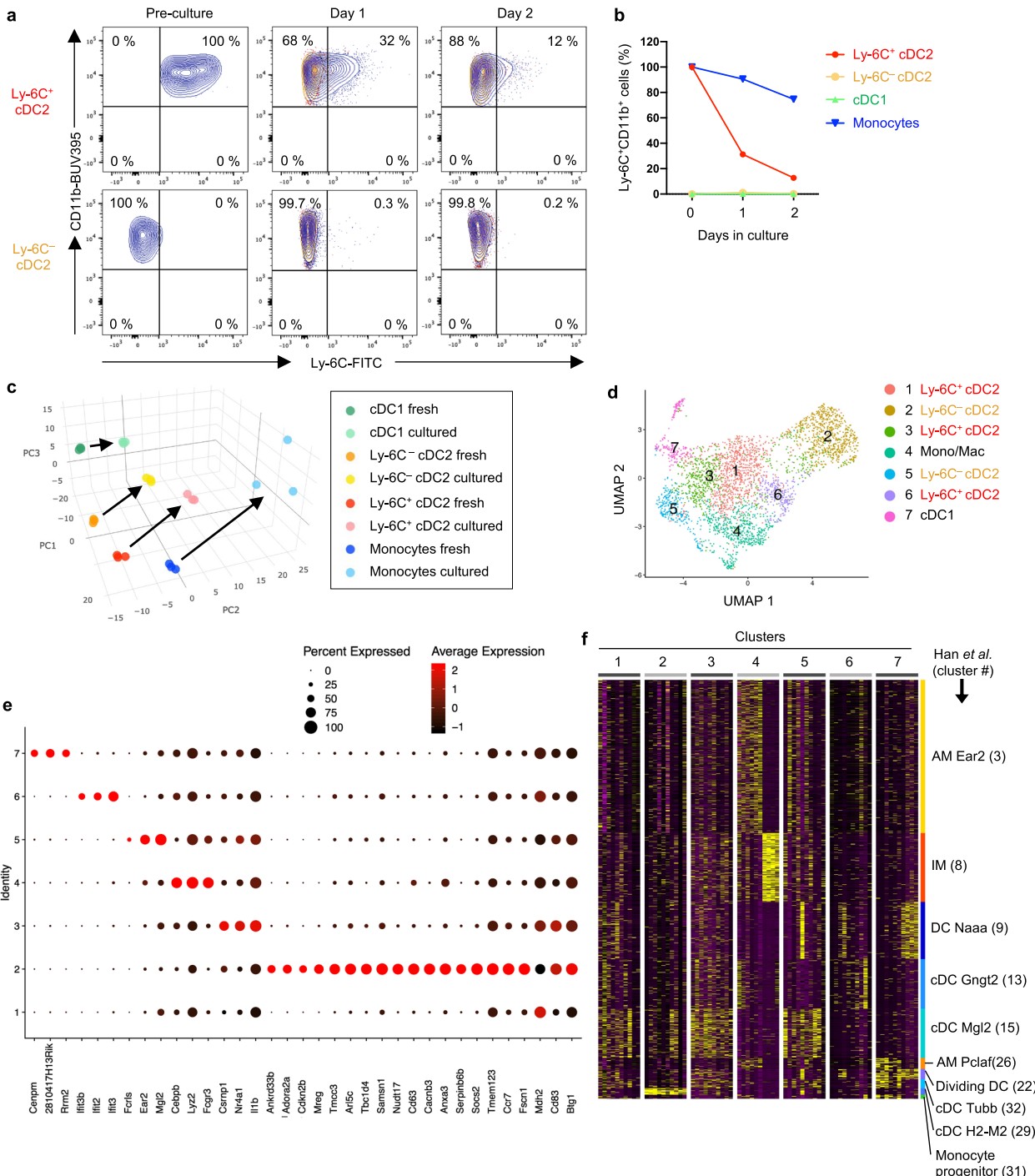

**Fig. 5 Developmental fate of Ly-6C⁺ cDC2. a**, **b** Surface Ly-6C on freshly isolated cDC2, and after their ex vivo culture. Representative cytograms for Ly-6C⁺ and Ly-6C⁻ cDC2 (**a**), and compiled data for Ly-6C⁺ cDC2, Ly-6C⁻ cDC2, cDC1 and monocytes (**b**) are shown. The gating strategy for cell sorting is shown in Supplementary Fig. 9a. Blue and orange contour plots show cells stained with anti-Ly-6C Abs and rat IgM isotype control, respectively. **c** PCA plots for gene expression (NanoString) in freshly isolated or cultured cDCs and monocytes. **d** UMAP plots showing cDC2 clusters identified by Seurat analysis of scRNA-Seq data. The key denotes identity of clusters, including five cDC2 clusters (three Ly-6C⁺ and two Ly-6C⁻). The gating strategy for cDC2 sorting is shown in Supplementary Fig. 3a. **e** Representative DEGs for the seven identified clusters. **f** Heatmap comparing expression of 10 diagnostic DEGs for each of the seven clusters shown in (**d**), with clusters identified by Han et al.[24]. Source data are provided as a Source Data file.

mediated sensitization (Fig. 6d). Unlike clusters 1, 3, and 6, this increase was not dependent on CCR2 and CX3CR1 (Supplementary Fig. 12a), suggesting that cluster 2 cells are derived from lung-resident cDCs, which are in cluster 5. To test this, we adoptively transferred Ly-6C⁻CD301b⁺CD200⁻ cDC2 from C57BL/6J (CD45.2) mouse lung into CD45.1 recipients. The

majority of donor-cDC2-derived CD45.2⁺ cells recovered from the recipient lungs at 1 day post-transfer had lost CD301b, but had undergone dramatic increases in CD200 (Fig. 7e and Supplementary Fig. 12c). Very few donor-derived cDC2 were recovered from recipients at 3 days post-transfer, but these cells also displayed the CD301b⁻Ly-6C⁻CD200⁺ phenotype (Fig. 7e

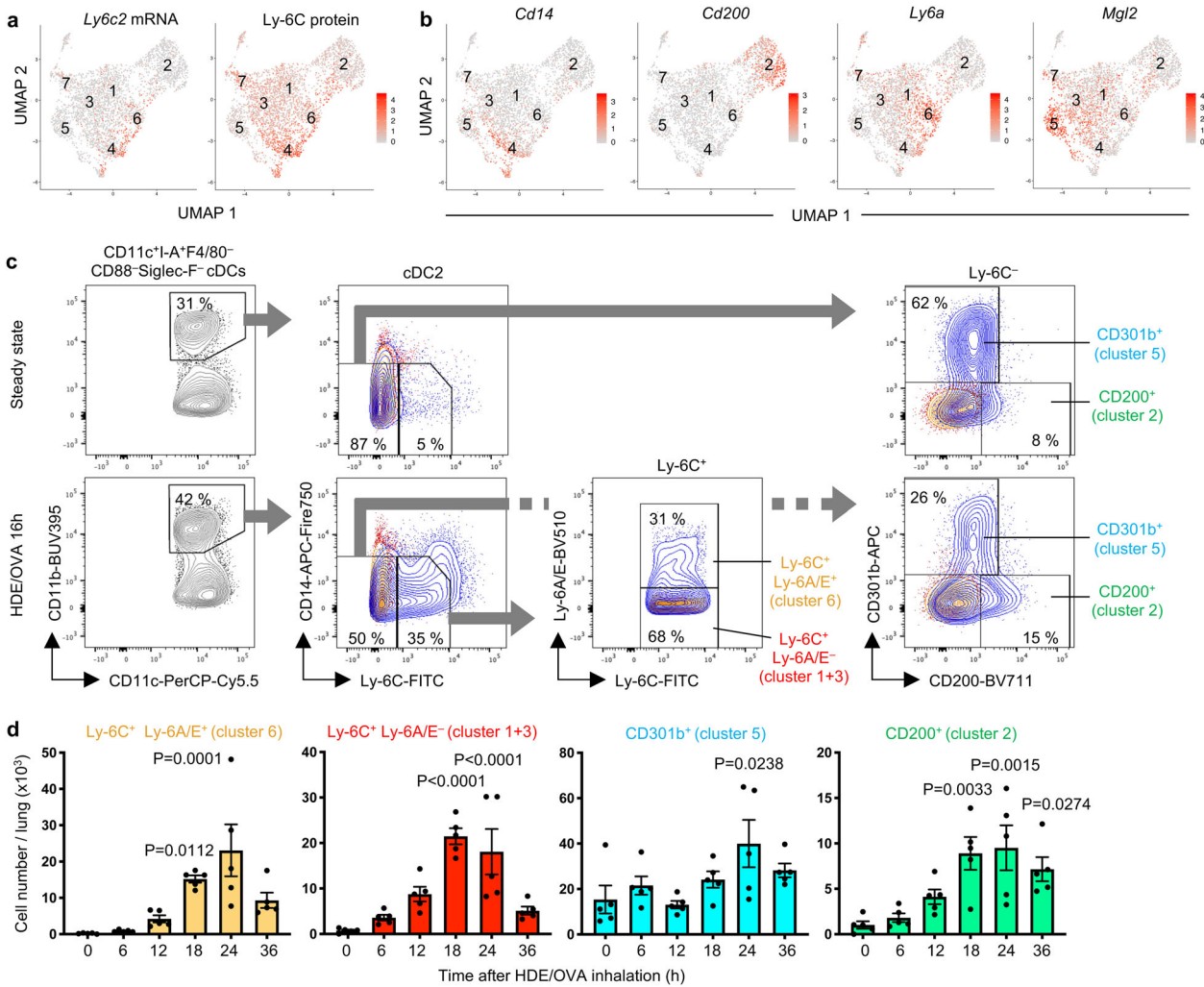

**Fig. 6 Surface markers and accumulation of CD11b$^+$ cDC2 subpopulations. a** UMAP plots showing *Ly6c2* gene expression and Ly-6C surface protein display, as identified by Seurat analysis of scRNA-Seq data. Expression levels are indicated by color differences. **b** UMAP plots showing unique expression of genes encoding cell surface proteins in various cDC2 clusters. **c** Flow cytometric gating strategy to resolve cDC2 (CD14$^{lo}$ cDCs) into four subpopulations: clusters 1 + 3 (Ly-6C$^+$Ly-6A/E$^-$), cluster 2 (Ly-6C$^-$CD200$^+$), cluster 5 (Ly-6C$^-$CD301b$^+$), and cluster 6 (Ly-6C$^+$Ly-6A/E$^+$) at steady state and 16 h after HDE/OVA inhalation. Additional gating information is shown in Supplementary Fig. 11a. Blue and orange contour plots show cells stained with specific Abs and isotype controls, respectively. **d** Time course for the number of DC2 clusters in the lung, as determined by flow cytometric analysis at steady state and at various time points following HDE/OVA inhalation. The gating strategy is shown in Supplementary Fig. 11b. Data were analyzed by ordinary one-way ANOVA with Sidak's multiple comparison test (*n* = 5). Data are presented as mean ± SEM. Statistical significance of each time point against steady state is shown. Source data are provided as a Source Data file.

and Supplementary Fig. 12c). These results demonstrate that CD301b$^+$cDC2 can give rise to fully mature CD200$^+$ cDC.

**Ly-6C$^+$ cDC2 subpopulation stimulates Th17 differentiation.** Previous studies have demonstrated that specific transcription factors are required in cDCs for their promotion of different T helper cell lineages; *Irf4* for Th2 and Th17; *Klf4* and *Relb* for Th2; and *Notch2* for Th17 responses[20,22,23,47–49]. Our scRNA-Seq analysis revealed selective expression of *Relb* in cluster 2, whereas *Irf4*, *Klf4*, and *Notch2* were not expressed in a cluster-specific manner (Fig. 8a). However, we did observe selective expression of *Il1b* gene in clusters 3 and 5 (Fig. 8a), suggesting these cDCs might preferentially stimulate Th17 differentiation. To test this experimentally, we purified four cDC2 subpopulations; cluster 1 + 3 (Ly-6C$^+$Ly-6A/E$^-$), cluster 2 (CD200$^+$), cluster 5 (CD301b$^+$), and cluster 6 (Ly-6C$^+$Ly-6A/E$^+$) from mouse lungs after HDE/OVA instillation, and separately co-cultured them with naive CD4$^+$ T cells from OT-II mice. Mature cDC2 in cluster 5 (CD301b$^+$) and

2 (CD200$^+$) potently induced T cell proliferation (Fig. 8b). Th17 differentiation was most strongly induced by cDC2 in cluster 1 + 3 (Ly-6C$^+$Ly-6A/E$^-$), followed by cluster 5 (CD301b$^+$) and cluster 6 (Ly-6C$^+$Ly-6A/E$^+$) (Fig. 8c and Supplementary Fig. 14b). cDC2 in cluster 2 (CD200$^+$) were the least effective in this regard. Cluster 2, while very poorly inducing Th17 differentiation, strongly induced Th2 differentiation. Cluster 5 had modest Th2-inducing activity, whereas the Ly-6C$^+$ subpopulations performed poorly in this regard (Fig. 8c). Taken together, the data indicate that partially mature cDC2 (mainly cluster 3) expressing *Il1b* preferentially induce Th17 differentiation, while fully mature cDC2 in cluster 2 selectively induce Th2 cells.

**Human counterparts of mouse lung cDC2.** We next investigated the relationship between the above-described clusters of mouse lung cDC2 and scRNA-Seq data recently reported for cells in human lung[50]. Human CD1c$^+$ cDC2 are heterogeneous and comprise at least two subpopulations (Supplementary Fig. 15a).

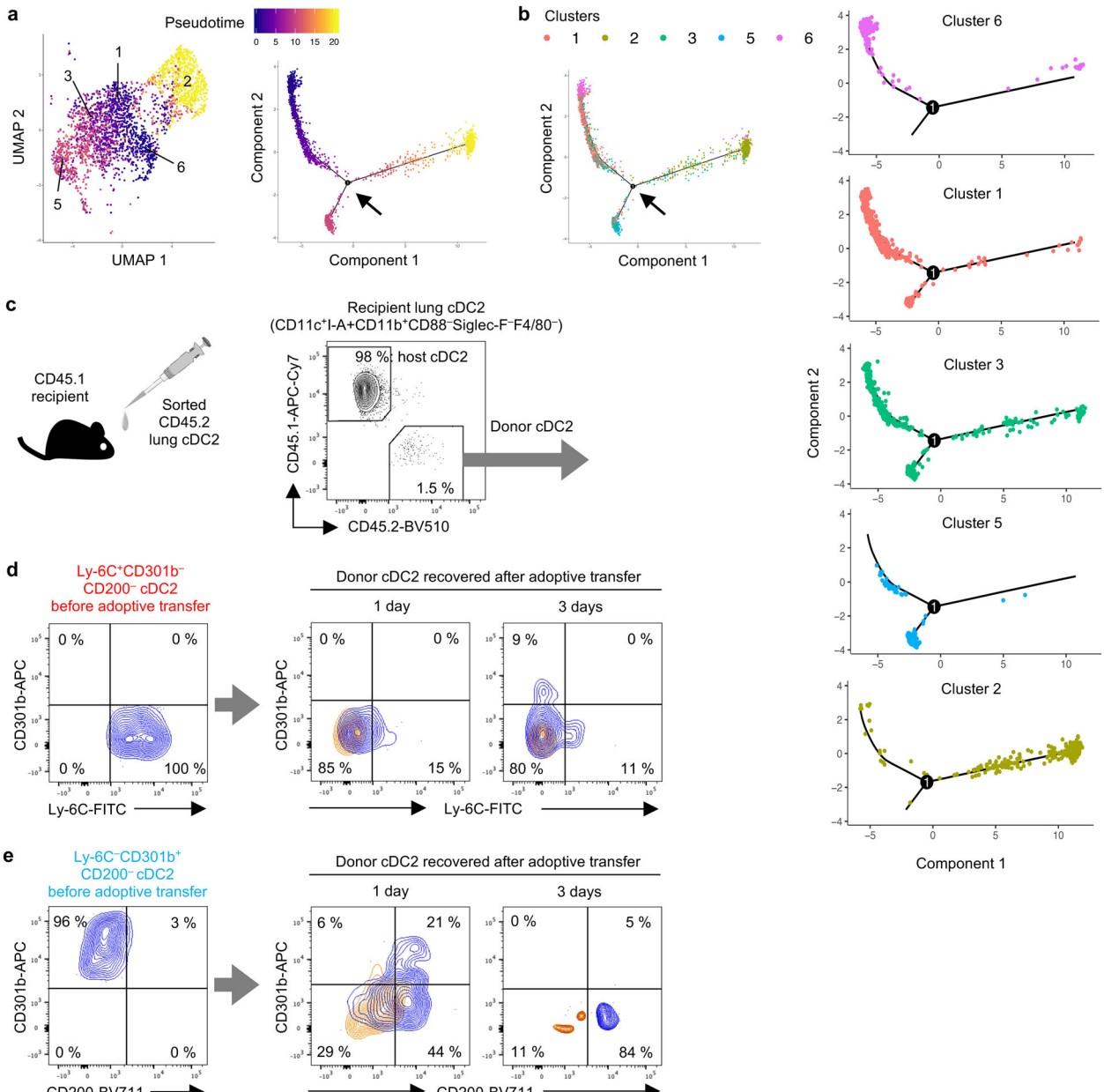

**Fig. 7 Maturation of cDC2 subpopulations. a** Pseudotime analysis (Monocle) of maturation pathways for five cDC2 clusters. A proposed developmental progression is indicated by different colors in the UMAP (left) or the trajectory plot (right) including a branching point for Ly-6C⁺ and Ly-6C⁻ cells. **b** Pseudotime-suggested developmental trajectory for five cDC2 clusters (1, 2, 3, 5, and 6). The clusters are color-coded, and a branching point (arrow) for Ly-6C⁺ and Ly-6C⁻ cells is indicated by an arrow or circled 1. A combination of all five clusters is shown (left panel), as well as trajectories for individual clusters (right panels). **c** Purified cDC2 subsets from C57BL/6 (CD45.2) were adoptively transferred to C57BL/6-CD45.1 mice. CD45.2⁺ donor-cDC2-derived cells were analyzed by flow cytometry. The gating strategies for cell sorting and analysis in flow cytometry are shown in Supplementary Fig. 13a, b. **d**, **e** Flow cytometric analysis of Ly-6C⁺CD301b⁻CD200⁻ (**d**) or Ly-6C⁻CD301b⁺CD200⁻ (**e**) before and 1 day and 3 days after adoptive transfer. Blue and orange counter plots show cells stained with specific Abs and isotype controls, respectively. A representative result from two independent experiments is shown.

To identify potential human counterparts of individual mouse cDC2 clusters, we evaluated human cell expression of DEGs that defined the various cDC2 clusters we had identified in the mouse lung. A human cDC2 subpopulation expressed *Ifitm1*, *Atf3*, and *Ccl17*, which were also highly expressed in mouse cDC2 clusters 1, 3, and 5 (Supplementary Fig. 15b). Noteworthy, this human cDC2 subpopulation also highly expressed *Il1b* (Supplementary Fig. 15a). By contrast, a different human cDC2 subpopulation had relatively low levels of *Il1b*. This subpopulation was similar to mouse cDC2 cluster 2, which also expressed *Marcksl*. These

results suggest that the former human cDC2 subpopulation might preferentially stimulate Th17 cell differentiation.

**Lung-resident cDCs induce Th17 cell differentiation**. Further analysis of our scRNA-Seq data revealed that expression of *Ccr7* is restricted to the Th2-inducing cDC2 in cluster 2 (Supplementary Fig. 10e). Because CCR7 is the chemokine receptor primarily responsible for directing migration of cDCs to lung-draining LNs[29], this finding suggests that most of the Th17-inducing cDC2 in clusters 1, 3, and 5 do not migrate to mLNs. To test this, we

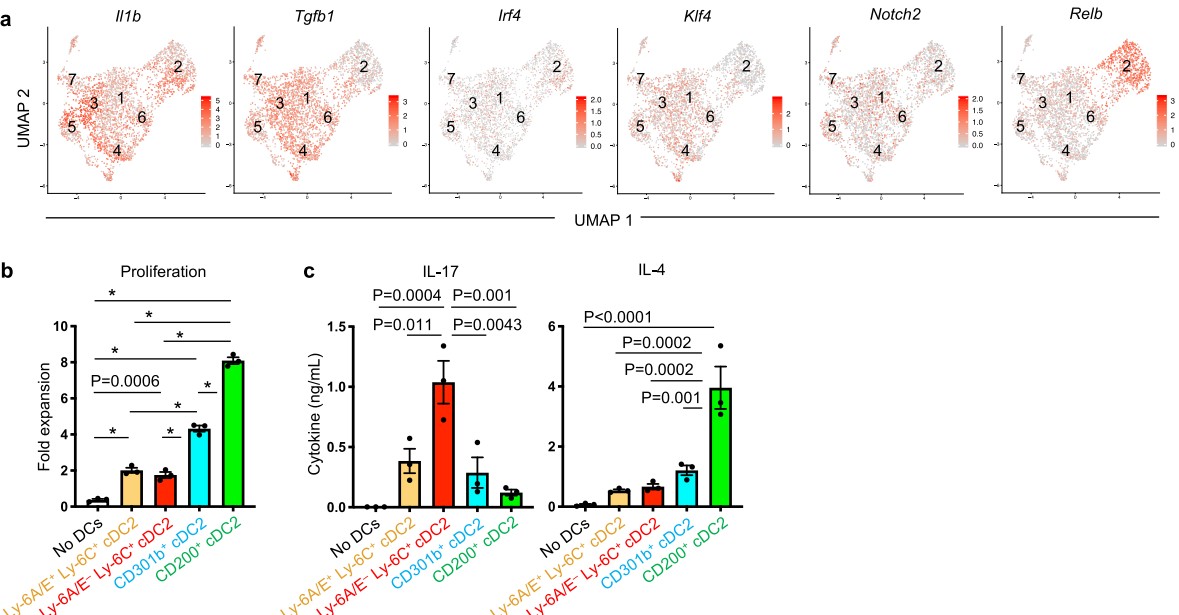

**Fig. 8 Select CD11b$^+$ cDC2 subpopulations stimulate Th17 differentiation. a** UMAP plots showing expression of genes encoding factors potentially regulating Th2 or Th17 cell differentiation in cDC2 clusters analyzed by scRNA-Seq. Expression levels are indicated by color difference and bars. **b, c** Proliferation (**b**) and IL-17 and IL-4 production (**c**) of OT-II CD4$^+$ T cells stimulated by the indicated cDC2 subpopulations. The gating strategy for cDC sorting is shown in Supplementary Fig. 14a. Data were analyzed by ordinary one-way ANOVA with Tukey's multiple comparison test ($n = 3$). $^*P < 0.0001$. Data are presented as mean ± SEM. Source data are provided as a Source Data file.

fluorescently labeled lung cDCs in vivo by instilling PKH26 dye into the airways of mice together with HDE/OVA, and analyzed PKH26$^+$ cDCs in lung-draining mLNs on the following day (Supplementary Fig. 16a). Among PKH26$^+$ migratory cDC2, more than 90% were CD200$^+$, including CD200 single positive and CD200$^+$CD301b$^+$ double positive cells (Supplementary Fig. 16a, b). By contrast, there were very few CD200$^-$ cells among PKH26$^+$ cDC2 in mLNs, consistent with their lack of CCR7. This raised the possibility that the latter cells, which promote Th17 differentiation ex vivo, might also fulfil this task in vivo as lung-resident cDCs. This would be consistent with our result that IL-17 production post-sensitization occurs in the lung with similar kinetics as production of that cytokine in lung-draining mLNs (Supplementary Fig. 1c). To confirm that Th17 cell differentiation can occur in the lung, we adoptively transferred IL-17 fate-mapping, OVA-specific (OT-II) CD4$^+$ T cells to recipient mice and evaluated Th17 cell development in the lung and mLNs at various times post-HDE/OVA sensitization[51]. The frequency of OVA-specific IL-17$^+$ Th17 cells among donor-derived CD4$^+$ T cells was higher in the lung than in mLNs at multiple time points post-HDE/OVA sensitization (Supplementary Fig. 16c, d). Our findings do not exclude the possibility of Th17 induction by mLN cDCs such as blood-derived cDCs, but clearly indicate that lung-resident cDCs can promote the development of allergen-specific Th17 cells.

To examine whether lung-resident cDCs are sufficient for the development of Th17-mediated neutrophilic allergic airway inflammation in vivo, we studied *Lymphotoxin alpha* (LTα)-deficient mice, which lack peripheral LNs[52]. Following splenectomy to avoid the confounding issue of sensitization in the spleen, animals were sensitized with LPS/OVA and subsequently challenged with aerosolized OVA (Supplementary Fig. 16e). WT mice developed eosinophilia, whereas *Lta*$^{-/-}$ mice did not (Supplementary Fig. 16f), suggesting that the spleen and/or LNs are necessary for Th2 cell development. By contrast, splenectomized *Lta*$^{-/-}$ mice developed

robust neutrophilia after challenge, demonstrating that the spleen and LNs are dispensable for Th17-dependent neutrophilia.

## Discussion

Identifying a consensus Th17-inducing cDCs in the lung has proven elusive, with different research groups reporting discrepant conclusions[18,22]. Reasoning that the heterogeneity of cDC2 might be responsible for these discrepancies, we deeply analyzed lung cDCs using a combination of high-dimensional mass cytometry, conventional flow cytometry, and scRNA-Seq. Mass cytometry revealed that a cDC2 subset with cell surface display of Ly-6C rapidly accumulates in the lung post-allergic sensitization, and flow cytometry-based cell sorting showed that these cDC2 can potently stimulate Th17 differentiation ex vivo. It is possible that this cDC subset has gone unappreciated until now because Ly-6C$^+$ cells are often assumed to be monocytes[20], and are therefore often excluded from cDC preparations. However, Ly-6C can be displayed by multiple cell types including preDCs[53]. Our use of indexed scRNA-Seq to simultaneously evaluate *Ly6c2* gene expression and Ly-6C protein suggested that immature cDC2 (cluster 6) express this gene, and that during maturation *Ly6c2* expression is lost first (clusters 1 and 3), followed by loss of Ly-6C protein (cluster 5). The immature status of cluster 6 is also suggested by its exclusive expression of *Ly6a*, which encodes the Ly-6A protein displayed on hematopoietic stem cells and DC progenitors[54,55]. This suggests an order of maturation from cluster 6 (Ly-6C$^+$Ly-6A/E$^+$) → clusters 1/3 (Ly-6C$^+$Ly-6A/E$^-$) → cluster 5 (CD301b$^+$), a path that is also consistent with the developmental order obtained from the pseudotime analysis, and in agreement with results of our adoptive transfer experiments with Ly-6C$^+$ cDC2. This developmental order might also explain the moderate Th17 promoting ability of Ly-6C$^-$ cDC2, as they likely arise from Ly-6C$^+$ cDC2 precursor cells. Although Ly-6C$^+$ cDC2 are not as abundant at steady state as they are post-HDE,

they are nonetheless present in the lung and might constitutively differentiate into the more mature CD301b⁺ lung-resident cDC2.

cDC2 in cluster 2 are almost absent at steady state, but their numbers increase dramatically following HDE/OVA inhalation. This accumulation was independent of CCR2 and CX3CR1, contrasting with Ly-6C⁺ cDC2 and preDCs, which are dependent on these chemokine receptors. This suggests that the cDC2 in cluster 2 arise from a pool of existing lung-resident preDCs or cDC2, but not from newly migrated preDCs, upon inhalation of agents such as HDE. Indeed, adoptive transfer experiment revealed that CD301b⁺ cDC2, which are the sole lung-resident cDC2 at steady state, mature to CD200⁺ cDC2, which are equivalent to the cells in cluster 2. These cDC2 were by far the most potent at promoting Th2 differentiation ex vivo, and they express many unique genes, including *Relb*, which encodes a transcription factor required in lung cDC for Th2 induction[49]. Thus, RELB might control the Th2-inducing function of these cDCs.

Another striking feature of cluster 2 in cDC2 is their exclusive expression of *Ccr7*. The limited expression of *Ccr7* among total cDC2 might explain why they are generally less migratory than cDC1[29]. It is not surprising that Ly-6C⁺ cDC2 (cluster 1, 3, and 6) do not express *Ccr7*, as they resemble immature or transitional cDCs, but the absence of appreciable *Ccr7* in mature cDCs within cluster 5 was unexpected. cDC2 in cluster 5 and their immediate precursors in cluster 3 are Th17-inducing lung-resident cDCs, as they express *Il1b* and *Tgfb1* and induce Th17 cell differentiation ex vivo. The function of lung-resident cDC2 is supported by our findings that Th17 responses (but not Th2 responses) can be detected in the lung shortly after HDE/OVA inhalation and that secondary lymphoid tissue is dispensable for Th17-dependent neutrophilia in HDE/OVA-mediated mouse model of asthma. In that model, we found that in addition to robust neutrophilia, lymphocyte numbers were also increased in the LN-deficient *Lta⁻/⁻* mice. This observation is consistent with previous studies showing that migration of cDCs from the lung to mLNs is required for the induction of tolerance to inhaled antigens[28,56]. MLNs might therefore be critical to both promote Th2 responses and regulate inflammatory responses. By contrast, mLNs are dispensable for Th17 differentiation, which can occur in the lung itself, and is likely driven by lung-resident cDCs.

Comparison of our scRNA-Seq data with previously published data allowed several parallels to be drawn. Han et al. identified multiple clusters of cDCs in the mouse lung at steady state[24], with an abundance of *Mgl2*⁺ cDC2. That cluster is transcriptionally most similar to our cluster 5, which was the major cDC2 cluster at steady state, and is somewhat similar to our clusters 1 and 3, which are developmentally related to cluster 5. *Gngt2*⁺ cDCs in the study of Han et al. are similar to the Ly-6C⁺ clusters 6 and 1 in our analysis but different from cluster 5, suggesting these cells are at an immature stage. Han et al. also detected a very minor population, *H2-M2*⁺ cluster 29, in lungs of naive mice, and this population is likely equivalent to cluster 2 in our analysis. Very recently, Bosteels et al. reported an analysis of cDC2 populations in the mouse lung following infection with the RSV-related pneumonia virus of mice (PVM)[25], and identified an inflammatory cDC2 population that has potent ability to induce proliferation and Th1 differentiation of CD4⁺ T cells. Comparing those data to our scRNA-Seq data revealed similarities between the inflammatory cDC2 reported by Bosteels et al. with cDC2 clusters 1 and 6 in our study. However, a counterpart of cluster 3, which preferentially stimulates Th17 cell differentiation, was not clearly seen in that comparison. Thus, depending on the stimulus to which they are exposed, cDCs might acquire different transcriptomic profiles and different functions.

Comparing our scRNA-Seq data with human lung cDC data reported by Vieira Braga et al.[50] revealed likely human counterparts of mouse cDC2 clusters. Mouse cDC2 in clusters 1, 3, and 5 in our study are likely analogous to a subpopulation of human cDC2 expressing *Il1b* reported by Vieira Braga et al. Recently Dutertre et al. reported that human CD5⁻CD163⁺CD14⁺ cDCs isolated from human blood potently stimulate Th17 differentiation[57]. We could not identify an unambiguous counterpart of that cDC cluster in our mouse cDC2 populations, possibly due to both species- and tissue-specific differences between the cells analyzed. However, it is possible that the *Il1b*-expressing human cDC2 cluster[50] overlaps with CD5⁻CD163⁺CD14⁺cDC2[57]. Finally, with regard to Th2 instruction, a subpopulation of human cDC2 that is distinct from *Il1b*⁺ cells expresses *Marcksl1*. We found that this gene is uniquely expressed by mouse cluster 2, suggesting that the *Il1b*⁻ *Marcksl1*⁺ human cDC2 subpopulation might be the human counterpart of the mouse cDC2 in cluster 2 (CD200⁺) that we observed to drive Th2 differentiation. Further characterization of both human and mouse cDCs will be necessary to test this.

In conclusion, we have identified multiple subpopulations of mouse cDC2, including a non-migratory lung-resident population that preferentially stimulates allergen-specific Th17 cell differentiation. These findings should assist in the development of both preventive and therapeutic strategies to control the induction and severity of Th17-mediated neutrophilic asthma.

## Methods

**Mice**. C57BL/6J, C57BL/6-*Batf3⁻/⁻* (B6.129S(C)-*Batf3*^tm1Kmm^/J), C57BL/6-*Ccr2⁻/⁻* (B6.129S4-*Ccr2*^tm1Ifc^/J), C57BL/6-*Cd11c*^Cre^ (B6.Cg-Tg(*Itgax*-cre)1-1*Reiz*/J), C57BL/6-CD45.1 (B6.SJL-*Ptprc*^a^ *Pepc*^b^/BoyJ), C57BL/6-*DTA*^fx^ (B6.129P2-Gt(ROSA) 26Sor^tm1(DTA)Lky^/J), C57BL/6-*Lta⁻/⁻* (B6.129S2-*Lta*^tm1Dch^/J), C57BL/6-OT-II TCR transgenic (B6.Cg(TcraTcrb)425Cbn/J), and C57BL/6-zDC-DTR (B6(Cg)-*Zbtb46*^tm1(HBEGF)Mnz^/J) mice were purchased from Jackson Laboratories. C57BL/6-*Flt3L⁻/⁻* (C57BL/6-*Flt3L*^tm1/mx^) and C57BL/6-*Cx3cr1⁻/⁻* (B6.129-*Cx3cr1*^tm1Zm^) mice were purchased from Taconic Biosciences (Germantown, NY, USA)[58]. OVA-specific *Il17a* fate-mapping mice on C57BL/6 background (B6.Cg-Il17a^tm1.1(EYFP/cre)Ehs^ Gt(ROSA)26Sor^tm9(CAG−tdTomato)Hze^ Tg(TcraTcrb)425Cbn) were generated as previously described[51]. *Ccr2⁻/⁻ Cx3cr1⁻/⁻* DKO mice were generated by crossing *Ccr2⁻/⁻* and *Cx3cr1⁻/⁻* mice[38]. ΔDC mice were generated by crossing *Cd11c*^Cre^ and *DTA*^fx^ mice[42]. *Batf3⁻/⁻* ΔDC mice were generated by crossing *Batf3⁻/⁻ Cd11c*^Cre^ and *Batf3⁻/⁻ DTA*^fx^ mice. Mice were bred and housed in specific pathogen-free conditions at the NIEHS with the following housing condition: light cycle: 7:00 a.m. to 7:00 p.m., temperature: 72 ± 2 °F, humidity: 40–60%. Mice were used between 6 and 12 weeks of age. All animal procedures complied with institutional guidelines were approved by the NIEHS Animal Care and Use Committee.

**Allergic sensitization and mouse model of asthma**. For allergic sensitization, mice were lightly anesthetized with isoflurane and given two oropharyngeal (o.p.) instillations, 1 week apart, of 100 µg LPS-free OVA (Worthington Biomedical) with 10 µL HDE or 100 ng LPS (Millipore Sigma) in a total volume of 50 µL in PBS (HDE/OVA or LPS/OVA)[30]. The HDE was prepared as previously described[32,59]. Briefly, vacuumed dust samples from homes in North Carolina were passed through a coarse sieve, then extracted at 100 mg/mL with PBS at 4 °C with overnight mild agitation. The samples were centrifuged to remove insoluble debris, and supernatants were sterilized by passage through a 0.22 µm filter (Millipore Sigma). Endotoxin concentration was 50 ng LPS/10 µL HDE, as measured by a Limulus Amebocyte Lysate assay (Lonza, catalog #50-648U). In some experiments, 100 µg Alexa Fluor (AF) 647-conjugated OVA or 10 µg DQ-OVA (ThermoFisher Scientific) was used to analyze antigen uptake[60]. In some experiments, DTX (20 ng/g body weight) (List Biological Laboratories) was injected into peritoneal cavity of zDC-DTA mice prior to airway sensitization[34]. For the full mouse model of asthma, male mice were sensitized twice by o.p. instillations of HDE/OVA or LPS/OVA, and challenged 1 week after the second sensitization by exposing them to an aerosol of 1% OVA (grade V, Millipore Sigma) in PBS on a single occasion for 1 h. Following euthanasia with intraperitoneal injection of sodium pentobarbital (Vortech Pharmaceuticals) at 48 h after challenge, bronchoalveolar lavage fluid (BALF) and lung tissue were collected. BALF leukocytes were spun onto glass slides using Cytospin centrifuge (ThermoFisher Scientific), and stained with hematoxylin and eosin prior to microscopic analysis. Lung tissues were incubated in complete RPMI1640 (ThermoFisher Scientific) containing 10% fetal bovine serum (FBS; characterized, Hyclone), 50 µM β-mercaptoethanol, penicillin, streptomycin, and 10 µg/mL OVA for 24 h. Cytokines in the supernatant were measured by ELISA.

**Isolation of DCs, preDCs, and T cells**. Lungs were harvested from untreated mice or 16 h after instillation of HDE/OVA unless specified. Lungs were perfused by PBS injection into the right ventricle. For DC preparation, minced tissues were digested with Liberase TM (100 μg/mL) (Roche), Collagenase XI (250 μg/mL), Hyaluronidase (1 mg/mL), and DNase I (200 μg/mL) (Sigma Aldrich) for 30 min (mass cytometry) or 60 min (flow cytometry) at 37 °C[60]. The reaction was stopped by the addition of EDTA (20 mM final concentration). A single-cell suspension was prepared by sieving the digested tissue through a 70 μm nylon strainer (BD Biosciences). To enrich DCs, low-density cells in the lung cells were collected by gradient centrifugation using 16% Nycodenz (Accurate Chemical), and then washed with PBS containing 0.5% bovine serum albumin and 2 mM EDTA. For T cell preparations, lung or mLNs were digested for 30 min, and single-cell suspension generated after passing through a 70 μm strainer. To enrich T cells, mononuclear cells were enriched by gradient centrifugation using Histopaque 1083 (Millipore Sigma). To isolate preDCs, bone marrow was collected from femurs, tibia, humeri and sternum bones, and red blood cells lysed with ACK buffer containing 0.15 M ammonium chloride and 1 mM potassium bicarbonate. Cells were passed through cell strainers, and mononuclear cells were enriched by gradient centrifugation using Histopaque 1083. PreDCs were enriched using an automated magnet-activated cell sorter (AutoMACS) (Miltenyi) by negative selection with the following biotinylated Abs obtained from BD Biosciences (BD), BioLegend (BL), or eBioscience/ThermoFisher Scientific (eBio): anti-mouse CD3ε (145-2C11, BD 553060; 0.5 μg/mL), CD11b (M1/70, BD 553309; 0.5 μg/mL), CD19 (6D5, BL 115504; 0.5 μg/mL), CD45R-B220 (RA3-6B2, e-bio 13-0452-85; 0.5 μg/mL), CD49b (DX5, BD 553856; 0.5 μg/mL), Ly-6A/E (D7, BD 557404; 0.5 μg/mL), Ly-6G (1A8, BL 127604; 0.5 μg/mL), and TER119 (TER-119, BD 553672; 0.5 μg/mL)[38]. cDCs and preDCs were purified by flow cytometric sorting.

**Flow cytometric analysis and sorting**. Cells were diluted to 1–2 × 10⁶/100 μL and incubated with a non-specific binding blocking reagent cocktail of anti-mouse CD16/CD32 Ab (2.4G2) (10% culture supernatant), 5% normal mouse, and 5% rat serum (Jackson ImmunoResearch)[61]. Cell surface antigens were stained with the following fluorochrome-conjugated Abs obtained from BD Biosciences (BD), BioLegend (BL), or eBioscience/ThermoFisher Scientific (eBio). BUV395-anti-mouse CD3e (145-2C11, BD 563565; 1 μg/mL), BV510-anti-mouse CD3e (145-2C11, BD 563024; 1 μg/mL), APC-anti-mouse CD4 (RM4-5, BL 100516; 1 μg/mL), BUV395-anti-mouse CD11b (M1/70, BD 563553; 1 μg/mL), Alexa Fluor 488-anti-mouse CD11c (N418, eBio 53-0114082; 2.5 μg/mL), PerCP-Cy5.5-anti-mouse CD11c (N418, eBio 45-0114-82; 1 μg/mL), APC-Fire750-anti-mouse CD14 (Sa14-2, BL 123332; 1 μg/mL), BV510-anti-mouse CD14 (Sa214-2, BL 123323; 1 μg/mL), BV510-anti-mouse CD19 (1D3, BD 562956; 1 μg/mL), APC-Cy7-anti-mouse CD45.1 (A20, BL 110716; 1 μg/mL), BV510-anti-mouse CD45.2 (104, BL 109837; 1 μg/mL), BV711-anti-mouse CD45.2 (104, BL 109847; 1 μg/mL), BV510-anti-mouse CD45R-B220 (RA3-6B2, BD 563103; 1 μg/mL), PE- anti-mouse CD80 (16-10A1, BL 104707; 0.5 μg/mL), PE- anti-mouse CD86 (GL1, eBio 12-0862-82; 0.5 μg/mL), APC-anti-mouse CD88 (20/70, BL 135808; 1 μg/mL), PE-anti-mouse CD88 (20/70, BL 135806; 0.5 μg/mL), PerCP-Cy5.5-anti-mouse CD88 (20/70, BL 135813; 1 μg/mL), BV510-anti-mouse CD103 (2E7, BL 121423; 1 μg/mL), APC-anti-mouse CD103 (2E7, eBio 17-1031-82; 1 μg/mL), PE-anti-mouse CD135 (A2F10, BL 135306; 1 μg/mL for bone marrow cells, 2 μg/mL for lung cells), FITC-anti-mouse CD172a (P84, BD 560316; 2.5 μg/mL), Alexa Fluor 647-anti-mouse CD200 (OX-90, BL 123816; 1.25 μg/mL), BV711-anti-mouse CD200 (OX-90, BD 745548; 1 μg/mL), PE-anti-mouse CD200 (OX-90, BL 123807; 0.5 μg/mL), APC-anti-mouse CD301b (URA-1, BL 146813; 1 μg/mL), PE-anti-mouse CD301b (URA-1, BL 146804; 1 μg/mL), PE-anti-mouse F4/80 (BM8, eBio 12-4801-82; 1 μg/mL), BUV737-anti-mouse F4/80 (T45-2342, BD 749283; 1 μg/mL), PE-Dazzle594-anti-mouse F4/80 (BM8, BL 123146; 1 μg/mL), BV510-anti-mouse Ly-6A/E (D7, BL 108129; 1 μg/mL), BV711-anti-mouse Ly-6A/E (D7, BL 108131; 0.5 μg/mL), APC-eFluor780-anti-mouse Ly-6C (HK1.4, eBio 47-5932-82; 1 μg/mL), BV711-anti-mouse Ly-6C (HK1.4, BL 128037; 1 μg/mL), FITC-anti-mouse Ly-6C (AL-21, BD 553104; 2.5 μg/mL), BV510-anti-mouse Ly-6G (1A8, BL 127633; 0.5 μg/mL), eFluor450-anti-mouse MHC class-II I-A[b] (AF6-120.1, eBio 48-5320-82; 1 μg/mL), PE-anti-mouse MHC class-II I-A/I-E (M5/114.15.2, BD 557000, lot 60577; 0.125 μg/mL), PE-anti-mouse Siglec-F (E50-2440, BD 552126; 0.5 μg/mL), AF647-anti-mouse Siglec-F (E50-2440, BD 562680; 1 μg/mL), PerCP-Cy5.5-anti-mouse Siglec-F (E50-2440, BD 565526; 1 μg/mL), BV510-anti-mouse TER119 (TER-119, BD 563995; 0.5 μg/mL), PE-hamster IgG (eBio 12-4914-81; 1 μg/mL), APC-rat IgG2a (eBR2a, eBio 17-4321-81; 1 μg/mL), BV711-rat IgG2a (RTK2758, BL 400551; 1 μg/mL), PE-rat IgG2a (eBR2a, eBio 12-4321-82; 0.5 μg/mL), PE-rat IgG2b (eB149/10H5, eBio 12-4031-82) (0.5 μg/mL), and FITC-rat IgM (R4-22, BD 553942; 2.5 μg/mL). Stained cells were analyzed on LSR-Fortessa flow cytometer (BD Biosciences), and the data analyzed using FACS Diva (BD Biosciences) and Cytobank (Cytobank) or FlowJo (Treestar) software. Only single cells were analyzed or purified, and dead cells stained with eFluor780-conjugated Live/Dead dye (ThermoFisher Scientific) were excluded from analysis. For purification, stained cells were sorted using a cell sorter FACS ARIA-II (BD Biosciences). The gating strategies are depicted in the Supplementary figures. In some experiments, purified cells were processed by Cytospin Cytocentrifuge (ThermoFisher Scientific) and photographed under a Zeiss AxioObserver Z1 microscope with Zen software (Carl Zeiss).

**Mass cytometry**. Low-density cells (3 × 10⁶) isolated from perfused lungs of C57BL/6 mice were incubated for 5 min with 1 μM of Cell-ID Cisplatin (Fluidigm) at room temperature to label dead cells, then washed with Maxpar Cell Staining Buffer (Fluidigm). To block non-specific Ab binding, cells were incubated with anti-mouse CD16/CD32 Abs, normal mouse serum, and rat serum, then incubated with 50 μL of metal-conjugated Abs (Supplementary Table 1) for 30 min. Staining of cells with fluoresce-conjugated primary Abs was followed by metal-conjugated secondary Abs (Supplementary Table 1). Cells were fixed in 2% paraformaldehyde for 60 min, washed with Maxpar Cell Staining Buffer, and then incubated in 125 nM Cell-ID Intercalator-Ir (Fluidigm) in Maxpar Fix and Perm Buffer (Fluidigm) overnight. On the following day, cells were washed and filtered through BelArt SP Flowmi cell strainers (Fisher Scientific), and analyzed on a Helios mass cytometer (Fluidigm) with CyTOF 6.7 software (Fluidigm). Data were analyzed using Cytobank platform (Cytobank).

**Detection of allergen-specific Th17 and Th2 cells**. Lymphocytes were isolated from LNs and spleens of donor mice, including OT-II and *Il17a* fate-mapping OT-II mice, by gradient centrifugation using Histopaque 1083. For adoptive transfers of cells, 10⁷ lymphocytes were injected into tail veins of recipient mice. On the following day, recipient mice were sensitized by o.p. aspiration of HDE/OVA as described above, and mLNs and lungs were collected at various times post-sensitization. MLN cells (1 × 10⁶ cells/200 μL/well) or lungs (all lobes/0.5 mL/well) were cultured in complete RPMI1640 containing 10 μg/mL OVA for 24 h (lung) or 48 h (mLN cells). IL-17 and IL-13 in the supernatants were measured using specific ELISA. In experiments using *Il17a* fate-mapping OT-II donor and CD45.1 recipient mice, donor CD4⁺ T cells were detected by staining with Abs against CD45.2, CD3ε, and CD4. Th17 cells were identified by TdTomato fluorescence in flow cytometry (Supplementary Fig. 2b).

**Co-culture of DCs and CD4⁺ T cells**. Naïve CD4⁺ T cells were purified from LNs and spleens by AutoMACS (Miltenyi) using streptavidin-conjugated MACS beads (Miltenyi) and a biotinylated Ab cocktail containing the following Abs obtained from BD Biosciences or BioLegend; anti-mouse CD8α (53-6.7, BD 553029; 0.5 μg/mL), CD8α (53-6.7, BD 553029; 0.5 μg/mL), CD8β (53-5.8, BD 553039; 0.5 μg/mL), CD11b (M1/70, BD 553309; 0.5 μg/mL), CD11c (HL3, BD 553800; 0.5 μg/mL), CD16/32 (2.4G2, BD 553143; 0.5 μg/mL), CD19 (6D5, BL 115504; 0.5 μg/mL), CD25 (PC61, BL 102004; 0.5 μg/mL), CD49b (DX5, BD 553856; 0.5 μg/mL), I-A[b] (AF6.120.1, BL 116404; 0.5 μg/mL), Ly-6C/G (RB6-8C5, BD 553125; 0.5 μg/mL), and CD44 (IM7, BL 103004; 0.05 μg/mL)[60]. Naïve CD4⁺ T cells (5 × 10⁴ cells/well) and lung cDCs (5 × 10³ cells/well) were co-cultured in a 7.5% CO₂ incubator for 5 days in 200 μL complete Iscove's modified Dulbecco's medium (IMDM) containing 10% FBS (certified, Invitrogen), 50 μM β-mercaptoethanol, penicillin and streptomycin in a 96-well U-bottom plate (BD Biosciences)[62]. In some experiments, neutralizing Abs against mouse IL-1β (B122, BL 503514) or IL-6 (MP5-20F3, BL 504512), or hamster IgG (HTK888, BL 400940) or rat IgG₁ (RTK2071, BL 400432) isotype control Abs (5 μg/mL) were added to co-cultures of cDCs and T cells. Culture supernatant was collected 3 days after culture, and IL-2 was measured by ELISA. Cells were harvested and washed 5 days after culture, and viable cells were counted using Luna-FL cell counter (Logos Biosystems). To elicit effector T cell responses, T cells were incubated (1 × 10⁵ cells/200 μL/well) for 24 h in a 96-well flat-bottom plate coated with Abs to mouse CD3e (145-2C11, BL 100331; 1 μg/mL) and CD28 (37.51, BL 102116; 1 μg/mL). Cytokines in the supernatant of incubated T cells were measured by ELISA using Multiskan Ascent plate reader with Ascent 2.6 software (Thermo Electron) or BioPlex Immunoassay (Bio-Rad, catalogue #171G5013M) according to manufacturer's instruction.

**Ex vivo DC maturation assay**. Lung cDCs or monocytes were purified by flow cytometry and cultured (2 × 10⁴ cells/200 μL/well) for up to 2 days in complete RPMI-10 in 96-well U-bottom plates. The phenotypes of cultured cells were analyzed by flow cytometry. Total RNA was isolated from freshly isolated or cultured cells using NucleoSpin RNA XS kit according to the manufacturer's instruction (Takara Bio, catalogue #740902.50). mRNA expression was examined using the NanoString platform utilizing the Mouse Myeloid Innate Immunity Panel v2 (NanoString Technologies) that measures 732 endogenous and 20 housekeeping RNAs. RNA expression was quantified on the nCounter Digital Analyzer. Data were adjusted utilizing the manufacturer's positive and negative experimental control probes as well as housekeeping genes with nSolver 4.0 software (NanoString Technologies). The data were further analyzed by Partek 7.0 (Partek) and R3.5.2 (R Foundation) software. The results (log₂) are presented in Supplementary Data 1.

**In vivo DC maturation assay**. Ly-6C⁺CD301b⁻CD200⁻ or Ly-6C⁻CD301b⁺CD200⁻ cDC2 from HDE/OVA-sensitized C57BL/6 mice (CD45.2) were purified by flow cytometry and adoptively transferred to syngeneic CD45.1 mice (0.5–1.5 × 10⁵ cells/recipient) by o.p. aspiration. Phenotype of CD45.2⁺ donor-cDC2-derived cells recovered from recipient lungs was analyzed by flow cytometry.

**Transcriptome analysis of lung APCs**. RNA from lung cDCs and monocytes and BM preDCs ($1–5 \times 10^5$ cells/sample) harvested from naive or HDE/OVA-treated C57BL/6 mice was isolated using NucleoSpin RNA XS kit. Unstranded RNA-Seq libraries with unique barcode adapters were constructed from total RNA using TruSeq RNA sample prep kit v2 (Illumina, catalogue #RS-122) according to the manufacturer's instructions. Multiplexed cDNA libraries (2 pM each sample) were sequenced by the NIEHS Epigenomics and DNA Sequencing Core Laboratory on a NovaSeq 6000 (Illumina) as single-end 76-mers. The data were processed using RTA version 3.3.3. Reads were filtered to retain only those with mean base quality score >20. Filtered reads were mapped to the mm10 reference genome via STAR version 2.5 (parameters–outMultimapperOrder Random–outSAMattrIHstart 0–outFilterType BySJout–alignSJoverhangMin 8–limitBAMsortRAM 55000000000–outSAMstrandField intronMotif–outFilterIntronMotifs RemoveNoncanonical)[63]. Counts per gene were determined by Subread feature-Counts v1.5.0-p1 (parameters: -s0) for a set of gene models defined by RefSeq transcripts as downloaded from the UCSC Table Browser (http://genome.ucsc.edu/cgi-bin/hgTables) as of September 5, 2017[64]. DEGs were identified via DESeq2 v1.14.1 at an FDR threshold of 0.05. Top 200 DEGs among cDCs and monocytes are presented in Supplementary Data 2. Depth tracks were generated by STAR version 2.5 (parameters–outWigType bedGraph–outWigStrand Unstranded–outWigNorm None) and subsequently normalized by size factors reported from DESeq2[65]. PCA coordinates were determined by R3.3.2 function "prcomp" using the 500 most-variant genes, with rlog-transformed scores as calculated by DESeq2. The RNA-Seq expression heatmap was generated by R function "heatmap.2" with row-scaling of rlog-transformed scores as calculated by DESeq2.

**Indexed single-cell RNA sequencing**. CD11b$^+$ cDC2 from HDE/OVA-treated C57BL/6 mice were purified by flow cytometry, and stained with Total A-Seq oligo-conjugated Abs against CD11b and Ly-6C according to manufacturer's instruction (BioLegend) (https://www.biolegend.com/en-us/totalseq)[45]. The cells were counted and examined for viability using a TC-20 cell counter (Bio-Rad). 7000 live cells at $7 \times 10^5$ cells/mL concentration with 98% viability were loaded into the single-cell ChIP followed by forming single-cell emulsion in Chromium Controller (10x Genomics). The cDNA for antibody-derived transcripts (ADT) and gene-derived transcripts was generated and amplified according to manufacturer's instructions (10x Genomics and BioLegend). The Gene Expression library and the ADT Library were prepared using the Chromium Single Cell 3′ v2 library and gel bead kit v2 (10x Genomics, catalogue #PN-120267) and additional reagents recommended in the protocol of Total A-seq (BioLegend). The two libraries were mixed at 10:1 molar ratio (Gene Expression library to ADT library) and sequenced by the NIEHS Epigenomics and DNA Sequencing Core Laboratory on NextSeq 500 (Illumina) with paired-end sequencing (Read 1:30; Read 2:100). The data were processed using RTA version 2.4.11. A total of $5.9 \times 10^8$ reads were obtained.

**Analysis of scRNA-Seq data**

*scRNA-Seq raw data processing*. Alignment, barcode assignment, and unique molecular identifier (UMI) counting was performed using Cell Ranger 3.0.1 and the "cellranger count" command. Alignment was performed with the mouse mm10-1.2.0 reference. The following feature libraries were included for antibody sequencing: Ly-6C (sequence: AAGTCGTGAGGCATG) and CD11b (sequence: TGAAGGCTCATTTGT). Outputs from filtered count matrices were used for subsequent analyses. From Cell Ranger an estimated 3891 cells, 144,076 mean reads per cells, and 2653 median genes per cell were recovered. 90% of reads mapped to genome and 97.7% barcodes were valid in antibody sequencing.

*scRNA-Seq dimensionality reduction and clustering*. Data from scRNA-Seq were processed using the Seurat v3.0 package in R version 3.6.2 (http://satijalab.org/seurat/)[43]. Data were filtered on characteristics for homogeneity, including number of features (high threshold: 4500; low threshold: 1000), total RNA counts (high threshold: 30,000; low threshold: 250), proportion cycling (high threshold: 0.06; low threshold: 0.01), and proportion of mitochondrial RNA (high threshold: 0.025; low threshold: 0.005). Data were normalized and scaled for number of RNA features, proportion cycling, and proportion of mitochondrial RNA. Normalized and scaled gene expression data were projected onto principal components (PCs). The first 30 PCs were used for non-linear dimensionality reduction using Uniform Manifold Approximation and Projection (UMAP)[66]. Gene expression and meta-data were visualized using this UMAP projection. Clustering was performed using the "FindNeighbors" (k.param = 50), followed by the "FindClusters" (resolution = 0.5) functions of the Seurat v3.0 package in R version 3.6.2[43]. Cluster marker genes from res.0.5 in Seurat (described above) were generated by the "FindAllMarkers" function (Supplementary Table 2). Heatmaps corresponding to scRNA-Seq data reported by Han et al.[24], Bosteels et al.[25], and Dutertre et al.[57] were generated with R3.3.2 package heatmap.2 using normalized expression scores from Seurat analysis of lung cells with assigned cell type annotation.

*scRNA-Seq pseudotime analysis*. Pseudotime analysis was performed using Monocle 2[46] according to instructions provided on GitHub (http://cole-trapnell-lab.github.io/

monocle-release/docs/). In short, filtered (Seurat) raw count matrices were subset on Seurat res.0.5 clusters 1, 2, 3, 5, 6 and used to infer cellular developmental trajectories. This information was projected onto two-dimensional space using "DDRTree". In addition, inferred pseudotime from Monocle 2 was projected onto UMAP dimensions 1 and 2 from Seurat (described above).

**Statistics**. Data are presented as mean ± SEM. Statistics to analyze differences among groups using Prizm software are indicated in figure legends. $P < 0.05$ was considered significant.

**Reporting summary**. Further information on research design is available in the Nature Research Reporting Summary linked to this article.

## Data availability
The NanoString data have been deposited in the Gene Expression Omnibus (GEO) (https://www.ncbi.nlm.nih.gov/geo/) under accession code GSE156763, and are also provided in Supplementary Data 1. The bulk RNA-Seq data of lung APCs in this study have been deposited in GEO under accession code GSE149778. DEGs in the bulk RNA-Seq analysis are provided in Supplementary Data 2. The scRNA-Seq data of CD11b$^+$ cDC2 have been deposited in GEO under accession code GSE156527. All other data supporting the findings of this study are available from the corresponding author upon request. Source data are provided with this paper.

## Code availability
Custom codes were not created for data analyses in this study. Analysis followed publicly available instructions from Seurat (http://satijalab.org/seurat/) and Monocle (http://cole-trapnell-lab.github.io/monocle-release/docs/). Any additional information required for the analysis of data in this manuscript is available from the authors upon request.

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

## Acknowledgements

We thank Xin Xu, Jason Malphurs, Jian-Liang Li, Guang Hu, and Gregory Solomon for help with RNAseq and scRNA-Seq; Kevin Gerrish and Rickie Fannin for help with NanoString; Maria Sifre and Carl Bortner for help with flow cytometry and cell sorting; Marie Iannone and Kevin Katen for help with mass cytometry; Erica Scappini for help with cell imaging; Ligon Perrow for help with mouse colony management; Florent Ginhoux and Charles-Antoine Dutertre (Singapore Immunology Network) for providing their human DC scRNA-Seq dataset; Timothy Thoner for assistance with experiments; and Prashant Rai and Jennifer Martinez (NIEHS) for critical reading of the manuscript. This work was supported by the Intramural Research Program of the National Institutes of Health, the National Institute of Environmental Health Sciences (ZIA ES102025-09).

## Author contributions

H.N. conceived of the project. H.N. and G.I. designed experiments. G.I., H.N., K.N., and G.S.W. performed experiments, and analyzed data. S.A.G., P.W.K., and M.B.F. analyzed RNA-Seq data. H.N. and D.N.C. wrote the manuscript. All authors contributed to discussion and review of the manuscript.

## Competing interests

The authors declare no competing interests.
