## [Peer Review File. · Nature Communications]

REVIEWER COMMENTS

Reviewer #1 (Remarks to the Author):

This study addresses an area of interest in respiratory immunology, with scope for advancing current understanding of DC subsets in the lung and promotion of Th17 responses using mouse models. Overall the idea has general plausibility, novelty and an extensive body of work has been performed utilising some nice models/platforms.

Major comments

Data across many of the figures are described as a representative result of 2 (or 3) independent experiments. Do the statistics (data analysis) shown in the figures relate to only that single particular representative experiment? - In all cases, all of the data (from the 2 or 3 replicate experiments) with statistical analyses should be shown for each figure.

Some of the data presented within the figures (which looks potentially interesting) is not really discussed- for example, in Fig 1d- was expression of other surface markers between the cDC2 subsets significantly different eg MHCII/CD80/CD86? Results shown in major figures should be discussed (albeit briefly).

Please list directly all the Ab/fluorochromes used and the full panel(s) used for different experiments- capacity to reproduce these findings/ integrate into other areas is an important component of publication. This would also help clarify the antigen uptake experiments (was the panel used/ gating strategy changed (Fig1) to accommodate the OVA-conjugate?). Can the authors comment on uptake of OVA v surface binding.

Please show all relevant appropriate controls for the cytometry work.

Please indicate in all appropriate figures that report on HDE-induced APCs, the time point of data collection, validation relevant to the time course.

Presumably, ExtFig1a is steady state-It would be appropriate to show a direct comparison with HDE/OVA demonstrating Ly6C subsets in CD11b+cDC2 cells and direct link with fig 1 flow cytometry.

Figure 1 c- why are lung APC at steady state not shown?

Authors could easily provide a list of the top (100-200) differentially expressed genes/p values between cell populations in addition to those they have selected to illustrate. In Ext fig 2b-should the axis be Ly-6C+ not Ly-6+? Please clarify why the data is normalised to expression levels in monocytes?

Further comments

Where applicable, statements within the introduction should be tailored to indicate particular reference

to humans/mouse or applicability to both.

The phenotype, as used in this study, to define cell populations should be included in results text (at least when first mentioned eg IM, APC, cDC etc). It would be helpful to include phenotypes in figure legends.

Extended data figure 1: should CD3e be included in the phenotype for APC as per legend description- Do these proportions of cells in the lung replicate other reports?

Can the authors please comment on how the gating strategies for flow v mass cytometry compare in regards final gated cell populations?- are the differences in the splits in cDC subset proportions meaningful?

It is great that visualising the cells is included- however, it is difficult to see figure 2a. were these cells cultured?

Were the cDC in Figure 4 gated differently to that used previously- there appear to be inconsistencies in the definitions of total cDC phenotype provided?

Was this a mixed sex mouse model?

Did the authors quantify the eosinophilic/neutrophilic responses?

How was bone marrow collected?

Were samples from mice pooled- how many (sorting / phenotyping) or was the data always derived from individual mouse samples- ?

Re presentation of the flow cytometry plots- these could potentially be improved, for example- adjust to display cells off the axes.

Can the authors comment on the differences in IL17 production by Ly6C+cDC2 across figure 3d?

Can the authors please check that the use of the terms Ly6C+ cDC2 and Ly6C+ APC as phenotype descriptors have been used correctly throughout. What is the definition of APC used here- does this include eosinophils?

In methods-isolation of DC- flow gating is referred to supp fig 2a-please change to appropriate.

Please clarify in figure 1f (and elsewhere)- what does cell number mean?- total? Per weight tissue?

Whole lung? How many cells (average per cell type) were purified for RNA extraction?

Authors should take care not to overstate interpretation of the presented data.

Reviewer #2 (Remarks to the Author):

The novelty of this paper is the emphasis on Th17 induction by a subset of cDC2s, along with the combined use of CyTOF and ssRNAseq to try and define 'new' cDC2 subsets. The CyTOF and ssRNAseq data are convincing though it is unclear why the authors adopted CyTOF first (very subjective) and then follow with ssRNAseq (very objective), rather than use ssRNAseq data from which to then design the most appropriate CyTOF panel.

Though an interesting paper, it doesn't refer to a key and relevant recent paper from Bart Lambrecht's lab (Bosteels et al Immunity 2020, 52: 1039) which was published in May this year that essentially (and very comprehensively) shows that cDC2s can acquire features of monocyte derived cells during inflammation (including HDM), using ssRNAseq and lots of functional experiments. This seems an odd

omission, given the phenotype the current authors are reporting. Furthermore, the discoveries don't go much beyond the subset descriptions in the Bosteels paper.

In contrast to the CyTOF/ssRNAseq, the functional data are novel (particularly sorted subset induction of IL-17 in responding T cells, and in vivo depletion strategies), though that blocking IL-1 and IL-6 leading to reduced Th17 induction is hardly surprising. Some of the in vivo model data are borderline and slightly over-emphasised (eg why is there no significant increase in IL-17 in the Batf3 delta DC mice in Fig 4f?).

The development data/points made in Fig 5 are less impressive as it is unclear whether things that change in culture is necessarily relevant to what might be happening in vivo.

So overall, a paper of the calibre suitable for publication, but fails to relate their 'subset defining' data to, and reference, the Bosteels paper, which reduces the novelty of their finding several cDC2 subsets in the lung. The authors should explain how their data extends Bosteels result.

Reviewer #3 (Remarks to the Author):

The goal of this manuscript was to better define the dendritic cells of the lung and specifically, to identify the subset which drive Th17 differentiation. They utilize mass cytometry and single cell RNA-sequencing to identify 5 clusters of cDC2s which are present in the lung after HDE inhalation. Further, they assess developmental progression of the cDC2s and the ability of these cells to promote differentiation of Th cells.

Overall, their findings are original and the work is convincing.

Major Concerns which may strengthen their conclusions include:

- The mechanism behind the reduced capacity of the Ly-6C+ APC to induce proliferation of OTII naïve cells should be explored. Are there significant changes in expression of co-stimulatory molecules or IL-2?
- While the pseudotime analysis is a bioinformatics approach to identifying the developmental trajectory, experimentally it may be key to determine if the Ly-6C+ cells are immature cDC2s, through traced with a lineage marker (i.e using Ly6C as a lineage marker) to determine if they are in fact maturing into the other clusters in vivo?
- Since priming and differentiation of Th cells is often thought to occur in lymphatics, the authors should explore the lung draining lymph nodes for these cells and not rely solely on their lack of CCR7 to conclude they do not migrate to the mLN. A tracking experiment could also investigate which of these subsets are moving into the mLN from the lung.

Minor Concerns which may help communicate to the readers include:

- Fig. 1a demonstrates that cDCs purified from the HDE/OVA-treated mice prime the development of Th17 and Th2 cells ex vivo. Do the mice also develop a mixed Th2/Th17 response in vivo following sensitization or after challenge? - These data are later present in the manuscript. Please consider whether some of the data in figure 8 is better served earlier in the manuscript to show relevance of the examination of cDC2 role in the lung for Th cell differentiation.
- In Figure 3, the flow cytometry assay demonstrates that Ly-6C+ cells can uptake antigen in Fig. 3a, b, but the text suggests that they can uptake and "present the antigen to T cells" (ln 138-9) the ability presenting antigen to T cells is not assessed until proliferation assays presented in Fig. 3d. Please modify

this sentence.

- As written, Line 262-267 are a little confusing. Is Il1b expression expressed in the cluster that expresses Iftm1, Aft3 and Ccl17 or the one that expresses Marcksl1 (similar to Cluster 2). If truly the 'former' the whole sentence would make more sense before the Marcksl1 results.
- Ly6C is also expressed on PMNs. What are the most distinguishing features between these Ly6C+ APC and PMNs?
- In lines 292-294, the authors should note that the splenectomized Lta^{-/-} AND splenectomized WT mice developed robust neutrophilia after challenge. But, they should also explain why only the the Lta^{-/-} have a robust lymphocytic infiltrate? Where are these lymphocytes coming from?

The statistical analyses appear appropriate and valid for their studies. They perform most experiments twice showing representative data in the figures.

The methods section is sufficient for the reader's understanding of the experimental processes and protocols.

Reviewer #1

Remarks to the Author. This study addresses an area of interest in respiratory immunology, with scope for advancing current understanding of DC subsets in the lung and promotion of Th17 responses using mouse models. Overall the idea has general plausibility, novelty and an extensive body of work has been performed utilizing some nice models/platforms.

Major comments

Comment. *Data across many of the figures are described as a representative result of 2 (or 3) independent experiments. Do the statistics (data analysis) shown in the figures relate to only that single particular representative experiment? - In all cases, all of the data (from the 2 or 3 replicate experiments) with statistical analyses should be shown for each figure.*

Response. We usually present representative results for each type of experiment and all present statistics for combined data of all such experiments. However, for some experiments, we were unable to do that because of variation between repeat experiments. In particular, levels of cytokines (IL-17) in co-cultures of T cells and cDCs vary considerably from experiment to experiment (Fig. 3d). Thus, for those experiments we present data for the repeat experiments in Supplementary Fig. 7g.

Comment. *Some of the data presented within the figures (which looks potentially interesting) is not really discussed- for example, in Fig 1d- was expression of other surface markers between the cDC2 subsets significantly different eg MHCII/CD80/CD86? Results shown in major figures should be discussed (albeit briefly).*

Response. We added a description regarding the differences on MHC-II and co-stimulatory molecules among cDC subsets (Lines 170-175).

Comment. *Please list directly all the Ab/fluorochromes used and the full panel(s) used for different experiments– capacity to reproduce these findings/ integrate into other areas is an important component of publication. This would also help clarify the antigen uptake experiments (was the panel used/ gating strategy changed (Fig1) to accommodate the OVA-conjugate?). Can the authors comment on uptake of OVA v surface binding.*

Response. Antibodies and fluorochromes along with gating strategy for each figure are shown in Supplementary Figures and the Reporting Summary. Details of flow cytometry analysis for OVA-uptake by DCs is added to Supplementary Figure 6a,b. As reviewer indicated, OVA bound to the the surface of cDCs could contribute to their fluorescence. We therefore confirmed the ability of antigen processing by cDCs using DQ-OVA, which fluoresces upon its digestion by intracellular proteases. The results from DQ-OVA test are presented in Supplementary Fig. 7a,b. Text in the Results section was also altered to include the results of this experiment (Lines 153-158).

Comment. *Please show all relevant appropriate controls for the cytometry work.*

Response. We added appropriate negative controls for flow cytometry analyses (Fig. 1f, 3a, 5a, 6c, 7d,e, Supplementary Fig. 7a, 9b, 12b,c, and 15a).

Comment. *Please indicate in all appropriate figures that report on HDE-induced APCs, the time point of data collection, validation relevant to the time course.*

Response. To clarify the time for harvesting lung after HDE/OVA treatment, we added the following sentence in the Methods section; 'Lungs were harvested from untreated mice or 16 h after instillation of HDE/OVA unless specified.' (Line 459).

Comment. *Presumably, Ext Fig1a is steady state-It would be appropriate to show a direct comparison with HDE/OVA demonstrating Ly6C subsets in CD11b+cDC2 cells and direct link with fig 1 flow cytometry. Figure 1 c- why are lung APC at steady state not shown?*

Response. A major advantage of mass cytometry is that more channels can be used, and more surface markers studied than in flow cytometry, thus enabling the identification of more cell populations. To benefit from mass cytometry, we used different gating strategy than we used for flow cytometry.

The data shown in former Fig. 1c was a different representation of the same (HDE/OVA) data shown in Fig. 1b, except the analysis was restricted to APCs. We did this to facilitate visual recognition of 2 distinct clusters in cDC2. However, based on this reviewer's comment, we might have made things more confusing. To simplify data presentation, without losing important data, we have removed the APC tSNE plot from the figure (now Fig. 1d).

Comment. *Authors could easily provide a list of the top (100-200) differentially expressed genes/p values between cell populations in addition to those they have selected to illustrate.*

Response. The top 200 differentially expressed genes among cDCs and monocytes, as determined by RNA-Seq analysis, are now presented in Supplementary Table 3 (Lines 576-577).

Comment. *In Ext fig 2b-should the axis be Ly-6C+ not Ly-6+? Please clarify why the data is normalised to expression levels in monocytes?*

Response. As the reviewer pointed out, the label should be 'Ly-6C+'. We apologize for this error. The label has been corrected in new Supplementary Fig. 4b. The purpose of this analysis was to compare monocytes with Ly-6C⁺ APCs to determine whether the latter are simply a subset of the former. For this reason, the data were normalized to monocytes.

Minor comments

Comment. *Where applicable, statements within the introduction should be tailored to indicate particular reference to humans/mouse or applicability to both.*

Response. Where appropriate, we now indicate mouse and/or human in the Introduction (Lines 59, 63-64, 66 and 67).

Comment. *The phenotype, as used in this study, to define cell populations should be included in results text (at least when first mentioned eg IM, APC, cDC etc). It would be helpful to include phenotypes in figure legends.*

Response. We added the phenotype (surface markers) of cDCs, AMs, IMs and monocytes in the Result section (Lines 107 and 113-116) and figure legends as follows.

APCs in mass cytometry: CD45⁺CD3 ϵ ⁻CD19⁻NK1.1⁻Ly-6G⁻ (legend for Supplementary Fig. 2b).

cDCs in flow cytometry: CD45⁺CD11c⁺I-A⁻CD88⁻Siglec-F⁻F4/80⁻ (legend for Supplementary Fig. 1d).

cDCs in mass cytometry: CD45⁺ CD11c⁺I-A⁺CD88⁻Siglec-F⁻F4/80⁻CD3 ϵ ⁻CD19⁻NK1.1⁻Ly-6G⁻ (legend for Supplementary Fig. 2a)

Monocytes in flow cytometry: CD45⁺CD11b⁺F4/80⁺CD88⁻Siglec-F⁻Live/Dead⁻ (legend for Supplementary Fig. 3c)

AMs in flow cytometry: CD45⁺Siglec-F⁺CD11c⁺ Live/Dead⁻ (legend for Supplementary Fig. 3c)

IMs in flow cytometry: CD45⁺CD88⁺F4/80⁺Siglec-F⁻Live/Dead⁻ (legend for Supplementary Fig. 3c)

Comment. *Extended data figure 1: should CD3e be included in the phenotype for APC as per legend description- Do these proportions of cells in the lung replicate other reports?*

Response. We thank the reviewer for pointing that out. CD3 ϵ was added to Supplementary Fig. 2a legend. The relative proportions of different cell populations obtained are dependent on the methods used for cell preparation. Such variables include enzyme types and concentrations, incubation times for tissue digestion, and methods for gradient centrifugation. For example, in our hands, 30 minutes of digestion is sufficient to release large numbers of lymphocytes, whereas 60 minutes of digestion is required to release large numbers of DCs. As the current study focused on cDCs, our method was optimized for those cells. For these reasons, it is difficult to directly compare cell proportions in different studies, but our results were consistent among multiple, independent experiments.

Comment. *Can the authors please comment on how the gating strategies for flow v mass*

cytometry compare in regards final gated cell populations?- are the differences in the splits in cDC subset proportions meaningful?

Response. A great advantage of mass cytometry is that many more cell surface markers can be studied than is possible with flow cytometry. For that reason, in the mass cytometry experiments, we were able to include markers of lymphocytes and granulocytes, in addition to those for DC and macrophages. Therefore, the gating strategies used for mass cytometry and flow cytometry were different. The multiparameter feature of mass cytometry allowed us to use unsupervised analyses, as represented in tSNE plots shown in Fig. 1d. This analysis allowed us to identify cDC2-like cells. Although fewer parameters can be used in flow cytometry, this approach works well for quantitation and purification of cells based on known display of specific surface markers (supervised analysis). Final gating and cell proportions can be similar between mass cytometry and flow cytometry, but their purposes are principally different. We changed a sentence about the mass cytometry for unsupervised analysis in the Result section (Lines 103-105).

Comment. *It is great that visualizing the cells is included- however, it is difficult to see figure 2a. were these cells cultured?*

Response. The images of cDCs and monocytes presented in Fig. 2a are of freshly isolated primary cells. Cells were purified by flow cytometry sorting, spun on glass slides using cytospin, and photographed under a microscope. We replaced the former images with those of a higher resolution (Fig. 2a). Procedures related to this have been added to the Methods section (Lines 494-496).

Comment. *Were the cDC in Figure 4 gated differently to that used previously- there appear to be inconsistencies in the definitions of total cDC phenotype provided?*

Response. Different fluorochromes were used in some experiments, depending on the availability of labeled antibodies. However, the phenotypes (gating strategies) of cDCs were principally same unless otherwise specified. We disclosed the gating strategy with fluorochrome we used for each channel. The gating strategy for Fig. 4a analysis is shown in Supplementary Fig. 3a. The surface markers of total cDCs (CD45⁺CD11c⁺I-A⁺CD88⁻F4/80⁻Siglec-F⁻Live/Dead⁻) is described in the legend for Fig. 4a.

Comment. *Was this a mixed sex mouse model?*

Response. Both sexes were used throughout the study. However, the same sex was used whenever we made comparisons between wildtype and genetically altered mice, or between different treatment groups. Males were used for mouse models of asthma (HDE/OVA sensitization followed by OVA challenge) (Fig. 4e,f). This information, as well as additional information relating to these experiments are provided in the Methods section as follows; 'For the full mouse model of asthma, male mice were sensitized twice by o.p. instillations of HDE/OVA or LPS/OVA, and

challenged 1 week after the second sensitization by exposing them to an aerosol of 1% OVA (grade V, Millipore Sigma) in PBS on a single occasion for 1 hour.' (Lines 447-450).

Comment. *Did the authors quantify the eosinophilic/neutrophilic responses?*

Response. We quantitated the number of eosinophils and neutrophils in allergic airway inflammation (Fig. 4f and Supplementary Fig. 1a,b, 15f). A change was made in a sentence (Lines 87-90).

Comment. *How was bone marrow collected?*

Response. To isolate preDCs, bone marrow was collected from femurs, tibia, humeri and sternum bones, and red blood cells lysed with ACK buffer containing 0.15 M ammonium chloride and 1 mM potassium bicarbonate. Cells were passed through cell strainers, and mononuclear cells were enriched by gradient centrifugation using Histopaque 1083. Pre-DCs were purified as described previously (Nakano *et al.*, *J Leukoc Biol* 2017). We added sentences to describe BM cell preparation in the Methods section (Lines 466-470).

Comment. *Were samples from mice pooled- how many (sorting / phenotyping) or was the data always derived from individual mouse samples- ?*

Response. For cell sorting, lung cells from multiple mice (6-18/sample) were pooled. For phenotyping analysis, lung or bone marrow cells from individual mice were analyzed unless otherwise specified.

Comment. *Re presentation of the flow cytometry plots- these could potentially be improved, for example- adjust to display cells off the axes.*

Response. We changed all cytograms to include display of cells formerly off the axis (Fig 1f, 4a,c, 5a, 6c, 7d,e, and Supplementary Fig. 1d, 2a, 3a,b, 5a,b,c, 6a,b, 7a,e 8a,b, 9a,b, 11a,b, 12 b,c, 13a,b, 15a,c and 16a,b,c).

Comment. *Can the authors comment on the differences in IL17 production by Ly6C+cDC2 across figure 3d?*

Response. The amount of cytokine production from cultured T cells can vary from experiment to experiment. This can be caused by the abundance of various factors in culture medium, including IL-2, and IFN- γ , and natural agonists of the aryl hydrocarbon receptor, (Veldhoen *et al.*, *J Exp Med* 2009, 206: 43-49). Th17 differentiation is also affected by the balance of cDC2 subpopulations, and contamination of cDC cultures with low numbers of cDC2-like monocytes, or other cells, is never exactly the same in two different experiments. Nonetheless, we reanalyzed IL-17 in the culture supernatants, and new graphs are now presented (Fig. 3e,f).

Comment. *Can the authors please check that the use of the terms Ly6C⁺ cDC2 and Ly6C⁺ APC as phenotype descriptors have been used correctly throughout. What is the definition of APC used here- does this include eosinophils?*

Response. Although eosinophils are not antigen-presenting cells, they were not excluded in our gating strategy for 'APCs' in mass cytometry in Supplementary Fig. 2a. Therefore, we describe cells as 'APCs + eosinophils' the figure. To avoid confusion, we added a description 'APCs gates (CD45⁺CD3e⁻CD19⁻NK1.1⁻Ly-6G⁻) including eosinophils' in the legend for Supplementary Fig. 2b.

Comment. *In methods-isolation of DC- flow gating is referred to supp fig 2a-please change to appropriate.*

Response. Gating strategies of flow cytometry are presented in Supplementary Figures. Different gating strategies including changes for fluorochrome-labeling of antibodies are depicted in those Supplementary Figures. The gating strategies corresponding to flow cytometry analyses are indicated in Figure legends (Fig. 1c,f,g, 2e,f, 3a, 4a,c, 5a, 6c,d, 7c-e, 8b, and Supplementary Fig. 3c, 5b, 7a,b, 8a,c, 12a,b,c and 15a,c).

Comment. *Please clarify in figure 1f (and elsewhere)- what does cell number mean?- total? Per weight tissue? Whole lung?*

Response. To clarify the Y-axis, we changed the label to 'Cell number/lung' in Fig. 1g, Fig. 2e,f, Fig. 4b,d, Fig. 6d and Supplementary Fig. 3c, 5d, 8b,d, 12a) and 'Cell number/mouse' in Fig. 4g and Supplementary Fig. 1b, 15f).

Comment. *How many cells (average per cell type) were purified for RNA extraction?*

Response. We used 1-5x10⁵ purified cells for RNA extraction in bulk RNA-seq analysis. We added the cell number and cDNA library concentration to the Methods section (Lines 562 and 566).

Comment. *Authors should take care not to overstate interpretation of the presented data.*

Response. We thank for reviewer's comment. To test the ability of antigen processing by cDCs, we added the results of an experiment using OVA-DQ (Supplementary Fig. 7a,b) and changed the sentence regarding antigen uptake as follows; 'The antigen processing ability of cDCs was also analyzed using DQ-OVA, which fluoresces upon its digestion by intracellular proteases. The frequency of OVA⁺ Ly-6C⁺ cDC2 and their mean fluorescent intensity (MFI) were either similar to, or greater than, corresponding values for the other cDC subsets (Supplementary Fig. 7a,b), suggesting that Ly-6C⁺ cDC2 have potential to capture

antigens, degrade them, and present antigen-derived peptides to T cells.' (Lines 153-158).

Reviewer #2 (Remarks to the Author):

Comment. *The novelty of this paper is the emphasis on Th17 induction by a subset of cDC2s, along with the combined use of CyTOF and ssRNAseq to try and define 'new' cDC2 subsets. The CyTOF and ssRNAseq data are convincing though it is unclear why the authors adopted CyTOF first (very subjective) and then follow with ssRNAseq (very objective), rather than use ssRNAseq data from which to then design the most appropriate CyTOF panel.*

Response. We thank for reviewer's comment. Our objective in the current study was to identify the cDC subset that promotes Th17 differentiation. We found Ly-6C to be an essential key surface marker for discriminating between Th17-inducing cDC2 from other cDC2 subsets, as shown in Fig. 3d, 8c and Supplementary Fig. 7g. Having this information on Ly-6C, we included Ly-6C protein in our scRNA-Seq analysis. If we analyzed cDC2 by scRNA-Seq first, we would not have been able to identify the Ly-6C protein-positive Ly6c2 mRNA-negative clusters (cluster 1 and 3), which are the main populations that induce Th17 differentiation. Therefore, presentation of our findings from mass cytometry, followed by scRNA-Seq was very important for accomplishing our objective in this study. To make this point clearer, we added a sentence 'Since Ly-6C is a surface marker on cDC2 that induce Th17 differentiation (Fig. 3d), we evaluated cell surface Ly-6C by labeling the cells prior to lysis with an oligonucleotide-labeled Ab directed at that protein (Indexed scRNA-Seq).' in the Results section (Lines 226-228).

Comment. *Though an interesting paper, it doesn't refer to a key and relevant recent paper from Bart Lambrecht's lab (Bosteels et al Immunity 2020, 52: 1039) which was published in May this year that essentially (and very comprehensively) shows that cDC2s can acquire features of monocyte derived cells during inflammation (including HDM), using ssRNAseq and lots of functional experiments. This seems an odd omission, given the phenotype the current authors are reporting. Furthermore, the discoveries don't go much beyond the subset descriptions in the Bosteels paper.*

Response. We thank for reviewer's comment regarding the paper reported by Bosteels et al (*Immunity* 2020, 52: 1039). As reviewer pointed, the authors analyzed cDC2 subsets including novel MAR-1⁺CD64⁺ inflammatory cDC2 using scRNA-Seq and flow cytometry, and reported that these cells induce Th1 differentiation. Their excellent study identified novel inflammatory cDC2, but Th17 induction by cDC subset was not clearly demonstrated, as MAR-1⁻ cDC2 induced both Th2 and Th17 differentiation. Interestingly, comparison of transcriptomic data for cDC2 clusters in our study with Bosteels' scRNA-Seq data revealed that the counterpart of cDC2 cluster 3, which is likely the main cDC population to induce Th17 cells, is not seen in Bosteels' analysis (Supplementary Fig. 10c). This

difference might be due to higher resolution in our unsupervised analysis than Bosteels' supervised clustering, or cDCs might acquire different transcriptomic profiles upon different stimuli. Nonetheless, as we succeeded in discriminating Th17-inducing cDC2 from Th2-inducing cDC2 in our present study, we believe our discoveries are important and warrant publication in *Nature Communications*. Moreover, as shown in Bosteels' paper, MAR-1 antibodies bind DCs through Fc portion but not through the epitope-specific binding site. Detection of cDC subsets using non-specific binding of antibody is not ideas, as other Fc receptor-positive cells can potentially be stained by the same antibodies. In our present study, we demonstrated specific surface markers to identify cDC2 subsets such as Ly-6C, Ly-6A/E, CD301b and CD200. Knowledge of those subset-specific surface markers will be beneficial to other researchers with an interest in lung cDC subsets. Thus, we believe our discoveries nicely build on Bosteels' paper. In the Results section of this revision, we have added a description regarding comparison between the published transcriptomes and our own (Lines 217-221). We also now discuss the similarities and differences in these data sets in our Discussion (Line 395-403).

Comment. *In contrast to the CyTOF/ssRNAseq, the functional data are novel (particularly sorted subset induction of IL-17 in responding T cells, and in vivo depletion strategies), though that blocking IL-1 and IL-6 leading to reduced Th17 induction is hardly surprising. Some of the in vivo model data are borderline and slightly over-emphasised (eg why is there no significant increase in IL-17 in the Batf3 delta DC mice in Fig 4f?).*

Response. We agree with reviewer's comments. The results from the experiment of IL-1 β and IL-6 blockade is predictable and not a novel mechanistic finding. However, we reasoned that for due diligence, we should verify that these cytokines are also important for Th17 induction by Ly-6C⁺ cDC2. The results are in agreement with previous reports and *I1b* expression by cDC2 cluster 3 identified in scRNA-Seq in the present study.

As reviewer indicated, IL-17 production by *Batf3*^{-/-} Δ DC mouse lungs was not significantly increased over that of wildtype mice. The animal model of airway inflammation using *Batf3*^{-/-} Δ DC mice might not be the optimal model to demonstrate the function of Th17-inducing cDC2, as the life span of Ly-6C⁺ cDC2 is short in these mice. A limitation in our experiment is the relatively small number of relatively mature Ly-6C⁺ cDC2 in *Batf3*^{-/-} Δ DC mice. However, as these mutant mice still produce IL-17 and develop neutrophilia in that model, in contrast to their inability to produce high amounts of IL-13 production and develop eosinophilia. We believe these findings make an important contribution to our understanding of how Th17 response develop in the lung.

Comment. *The development data/points made in Fig 5 are less impressive as it is unclear whether things that change in culture is necessarily relevant to what might be happening in vivo.*

Response. We thank for reviewer's comment. We reanalyzed Ly-6C display on cDC subsets and monocytes, and present these new results in Fig. 5a,b and Supplementary Fig. 9b. We have also made a change in the text to reflect this (Lines 206-208).

In addition, we tested maturation of cDC2 subpopulations *in vivo* after adoptive transfer. Ly-6C⁺ cDC2 lost Ly-6C from their surface and gained some CD301b. CD301b⁺ cDC2 had reduced CD301b but gained CD200 display on their surface, as now presented in Fig. 7d,e and Supplementary Fig. 12b,c. We also modified text in the manuscript to reflect these new results (Lines 255-263 and 268-275).

Comment. *So overall, a paper of the calibre suitable for publication, but fails to relate their 'subset defining' data to, and reference, the Bosteels paper, which reduces the novelty of their finding several cDC2 subsets in the lung. The authors should explain how their data extends Bosteels result.*

Response. As indicated above, we identified Th17-inducing cDC2 subsets, which was not clearly demonstrated by Bosteels *et al* (*Immunity* 2020, 52: 1039). Also, we found specific surface markers to identify cDC2 subsets such as Ly-6C, Ly-6A/E, CD301b and CD200. Identification of those subset-specific surface markers will be of benefit to many researchers studying lung DC biology. Thus, we believe our discoveries build and extend the data in Bosteels' paper.

Reviewer #3 (Remarks to the Author):

Comment. *The goal of this manuscript was to better define the dendritic cells of the lung and specifically, to identify the subset which drive Th17 differentiation. They utilize mass cytometry and single cell RNA-sequencing to identify 5 clusters of cDC2s which are present in the lung after HDE inhalation. Further, they assess developmental progression of the cDC2s and the ability of these cells to promote differentiation of Th cells. Overall, their findings are original and the work is convincing.*

Response. We thank for reviewer's insightful and productive comments.

Major Concerns which may strength their conclusions include:

Comment. *The mechanism behind the reduced capacity of the Ly-6C⁺ APC to induce proliferation of OTII naïve cells should be explored. Are there significant changes in expression of co-stimulatory molecules or IL-2?*

Response. Mass cytometry results shown in Fig. 1e (Fig. 1d in previous version) indicate that HDE-induced APCs (Ly-6C⁺ cDC2) display relatively lower levels of CD40, CD80, CD86 and MHC-II on their surface compared with Ly-6C⁻ cDC2 or cDC1. Since MHC-II and co-stimulatory molecules may influence the ability to stimulate T cells, we tested the display levels of CD80, CD86 and MHC-II I-A/I-E

by flow cytometry, and verified lower levels of CD86 and MHC-II I-A/I-E on Ly-6C⁺ cDC2 (Supplementary Fig. 7e,f).

We also measured IL-2 levels in the culture supernatant of lung cDCs co-cultured with OT-II CD4⁺ T cells. IL-2 production induced by Ly-6C⁺ cDC2 was lower than that seen with Ly-6C⁻ cDC2 or cDC1 (Supplementary Fig. 7d). We added sentences to the Results section to reflect this (Lines 170-175).

Comment. *While the pseudotime analysis is a bioinformatics approach to identifying the developmental trajectory, experimentally it may be key to determine if the Ly-6C⁺ cells are immature cDC2s, through traced with a lineage marker (i.e using Ly6C as a lineage marker) to determine if they are in fact maturing into the other clusters in vivo?*

Response. We thank the reviewer for this excellent suggestion. In response to it, we attempted to study cDC2 maturation *in vivo* following adoptive transfer of purified Ly-6C⁺ (CD301b⁻CD200⁻) cDC2 and CD301b⁺ (Ly-6C⁻CD200⁻) cDC2 (Fig. 7c). We found that donor Ly-6C⁺ cDC2-derived cells lost Ly-6C at day 1, and some cells gained CD301b at day 3 post-transfer (Fig. 7d and Supplementary Fig. 12b). We also attempted to analyze the phenotype of donor Ly-6C⁺ cDC2-derived cells at a later time point. As shown in the cytogram below, the majority (62%) of donor-derived CD45.2⁺ cDC2 displayed Ly-6C⁻CD301b⁺ phenotype 8 days post-transfer, in agreement with upregulation of CD301b in some cells at day 3 (Fig.7d). However, as the number of CD45.2⁺ cDC2 recovered at day 8 was very low, we do not feel that these data should be presented in the manuscript.

Donor CD301b⁺ cDC2-derived cells displayed diminished CD301b, and gained CD200 (Fig. 7e and Supplementary Fig. 12c). The above results suggest that CD301b⁺ cells in cluster 5 descend from Ly-6C⁺ cells in clusters 6, 1 and 3, and that CD301b⁺ cDC2 can give rise to fully mature CD200⁺ cDC2.

Since these additional data demonstrated that cDC2 subpopulations are in common lineage, we revised the title, and changed text in the Abstract (Lines 35-41), the Result section (Lines 255-263 and 268-275) and the Discussion (Line 354-357).

Comment. *Since priming and differentiation of Th cells is often thought to occur in lymphatics, the authors should explore the lung draining lymph nodes for these cells and not rely solely on their lack of CCR7 to conclude they do not migrate to the mLN. A tracking experiment could also investigate which of these subsets are moving into the mLN from the lung.*

Response. Along with reviewer's suggestion, we tested the migration of lung cDC2 subpopulations by detecting PKH26⁺ cDCs in the mediastinal lymph nodes by flow cytometry after PKH26 dye instillation together with HDE/OVA to the airway. Among PKH26⁺ migratory cDC2, more than 90% cells were CD200⁺ including CD200⁻ single positive and CD200⁺CD301b⁺ double positive cells (Supplementary Fig. 15a,b). By contrast, there were very few CD200⁻ cells consistent with their lack of CCR7. We have also added sentences in the Result section (Lines 313-318).

Minor Concerns which may help communicate to the readers include:

Comment. *Fig. 1a demonstrates that cDCs purified from the HDE/OVA-treated mice prime the development of Th17 and Th2 cells ex vivo. Do the mice also develop a mixed Th2/Th17 response in vivo following sensitization or after challenge? - These data are later present in the manuscript. Please consider whether some of the data in figure 8 is better served earlier in the manuscript to show relevance of the examination of cDC2 role in the lung for Th cell differentiation.*

Response. We thank the reviewer for this suggestion. We agree that presentation of Th2 and Th17 responses in the mouse model of allergic airway inflammation at the beginning of manuscript might be better. Accordingly, we added a diagram of mouse model experiment, and show data for IL-13 and IL-17, as well as neutrophilia and eosinophilia, in the mouse airway following HDE/OVA sensitization and OVA challenge (Fig. 1a,b and Supplementary 1a,b).

Also, we moved the graphs presenting time course analyses of IL-13 (Th2) and IL-17 (Th17) in the lung and mediastinal lymph nodes to Supplementary Fig. 1c. We feel this change makes for a better transition and helps to show the relevance of subsequent experiments to address lung cDC2 roles for T cell differentiation.

Along with the above changes, we also changed text in the Result section (Lines 87-100).

Comment. *In Figure 3, the flow cytometry assay demonstrates that Ly-6C⁺ cells can uptake antigen in Fig. 3a, b, but the text suggests that they can uptake and "present the antigen to T cells" (In 138-9) the ability presenting antigen to T cells is not assessed until proliferation assays presented in Fig. 3d. Please modify this sentence.*

Response. We thank the reviewer for this suggestion. In our revision we now show antigen processing by cDCs using DQ-OVA, which fluoresces upon its digestion by intracellular proteases. The results from DQ-OVA test are presented in Supplementary Fig. 7a,b. Sentences in the Results were also altered to include this experiment (Lines 153-158). We state that these data suggest, 'Ly-6C⁺ cDC2 have potential to capture antigens, degrade them, and present antigen-derived

peptides to T cells.’. Subsequent cDC and T cell co-culture experiments confirm that these cDCs can indeed stimulate the proliferation of naïve CD4 T cells.

Comment. *As written, Line 262-267 are a little confusing. Is *Il1b* expression expressed in the cluster that expresses *Ifitm1*, *Aft3* and *Ccl17* or the one that expresses *Marcks11* (similar to Cluster 2). If truly the ‘former’ the whole sentence would make more sense before the *Marcks11* results.*

Response. We thank reviewer’s suggestion. To avoid confusion, we have changed the sentence be as follow;

‘A human cDC2 subpopulation expressed *Ifitm1*, *Aft3* and *Ccl17*, which were highly expressed in mouse cDC2 clusters 1, 3 and 5 (Supplementary Fig. 14b). Noteworthy, this human cDC2 subpopulation also highly expressed *Il1b* (Ext. Data Fig. 14a). By contrast, another human cDC2 subpopulation expressed *Marcks11*, which was highly expressed in mouse cDC2 cluster 2, and expressed relatively lower level of *Il1b*. These results suggest that the former human cDC2 subpopulation might preferentially stimulate Th17 cell differentiation.’ (Lines 301-307).

Comment. *Ly6C is also expressed on PMNs. What are the most distinguishing features between these Ly6C+ APC and PMNs?*

Response. Ly-6C APCs are MHC-II-positive, while PMNs are CD88⁺ and MHC-II⁻. To clarify the exclusion of neutrophils from our analyses, we added a sentence to describe gating strategy and surface markers (Lines 113-116). The gating strategies for cDC analyses are also shown in Supplementary Fig. 1d and other Supplementary Figures.

Comment. In lines 292-294, the authors should note that the splenectomized *Lta*^{-/-} AND splenectomized WT mice developed robust neutrophilia after challenge. But, they should also explain why only the the *Lta*^{-/-} have a robust lymphocytic infiltrate? Where are these lymphocytes coming from?

Response. This is insightful question. We do not yet fully understand the mechanisms leading to increased lymphocytes in *Lta*^{-/-} mice, but we respectfully suggest it is beyond the scope of the present manuscript to further address this experimentally. Nonetheless, to respond to the reviewer, we have revised the Discussion (Line 381-387) to include a description of this finding and a possible explanation for it. Briefly, previous studies have shown that CCR7-dependent migration of DCs from the lung to regional lymph nodes is required for the induction of tolerance to inhaled antigens. As *Lta*^{-/-} mice lack lymph nodes, these mice also lack that regulatory arm of the immune response, and might not be able to control lymphocyte proliferation. This is a plausible mechanism to explain increased numbers of lymphocytes in lungs of *Lta*^{-/-} mice.

Comment. The statistical analyses appear appropriate and valid for their studies. They perform most experiments twice showing representative data in the figures.

Response. We thank reviewer's positive comments.

Comment. *The methods section is sufficient for the reader's understanding of the experimental processes and protocols.*

Response. We thank reviewer's positive comments.

REVIEWERS' COMMENTS

Reviewer #1 (Remarks to the Author):

Most points have been addressed adequately-the paper has been significantly improved.

Reviewer #2 (Remarks to the Author):

I have re-read the manuscript and noted the additions made by the authors. They now clearly distinguish their research from that previously published by Bart Lambrecht's group and include an analysis within the manuscripts comparing the two datasets. I note that there are differences between the DC subsets in the two manuscripts that the authors put down to increased resolution of the data in their approach. The manuscript has been further improved in response to the other reviewers comments

Reviewer #3 (Remarks to the Author):

The goal of this manuscript was to better define the dendritic cells of the lung and specifically, to identify the subset which drive Th17 differentiation. They utilize mass cytometry and single cell RNA-sequencing to identify 5 clusters of cDC2s which are present in the lung after HDE inhalation. Further, they assess developmental progression of the cDC2s and the ability of these cells to promote differentiation of Th cells.

With this revision, they have:

1. demonstrated that cDC2 subpopulations are in common lineage
2. investigated the migration of lung cDC2 subpopulations
3. highlighted the novelty of their work beyond some recent publications
4. clarified several gating strategies and methods (i.e around eosinophils and PMNs)
5. corrected some minor errors/communications re:methods and data presentation.

I have no further comments at this time.